
# Review of fragility analyses for major building types in China with new implications for intensity-PGA relation development

Danhua Xin[1]*, James Edward Daniell[1,2], Friedemann Wenzel[1]

[1]Center for Disaster Management and Risk Reduction Technology (CEDIM) and Geophysical Institute, Karlsruhe Institute of Technology, Hertzstrasse 16, 76187, Karlsruhe, Germany

[2]General Sir John Monash Scholar, The General Sir John Monash Foundation, Level 5, 30 Collins Street, Melbourne,Victoria, 3000, Australia

*Correspondence to*: Danhua Xin (danhua.xin@kit.edu)

**Abstract**. The evaluation of the seismic fragility of buildings is one key task of earthquake safety and loss assessment. Many research reports and papers have been published over the past four decades that deal with the vulnerability of buildings to ground motion caused by earthquakes in China. We first scrutinized 69 papers and theses studying building damage for earthquakes occurred in densely populated areas. They represent observations where macro-seismic intensities have been determined according to the Chinese Official Seismic Intensity Scale. From these many studies we derived the median fragility functions (dependent on intensity) for four damage limit states of two most widely distributed building types: masonry and reinforced concrete. We also inspected 18 publications that provide analytical fragility functions (dependent on PGA) for the same damage classes and building categories. Thus, a solid fragility database based on both intensity and PGA is established for seismic prone areas in mainland China. A comprehensive view of the problems posed by the evaluation of fragility for different building types is given. Necessary comparison with international projects with similar focus is conducted. Based on the newly collected fragility database, we propose a new approach in deriving intensity-PGA relation by using fragility as the bridge and reasonable intensity-PGA relations are developed. This novel approach may shed light on new thought in decreasing the scatter in traditional intensity-PGA relation development, i.e., by further classifying observed macro-seismic intensities and instrumental ground motions based on difference in building seismic resistance capability.

## 1 Introduction

Field surveys after major disastrous earthquakes have shown that poor performance of buildings in earthquake affected areas is the leading cause of human fatalities and economic losses (Yuan, 2008). The evaluation of seismic fragility for existing building stocks has become a crucial issue due to the frequent occurrence of earthquakes in the last decades (Rota et al., 2010). Building fragility curves, defined as expected probability of exceeding specific building damage state under given earthquake ground shaking, have been developed for different typologies of buildings. They are required for the estimation of fatalities and monetary losses due to building structural damage. The development of fragility curves can be divided mainly into two approaches: empirical methods and analytical methods. Empirical methods are based on post-earthquake surveys for groups of buildings and considered to be the most reliable source, because they are directly correlated to the actual seismic behaviour of buildings (Maio and Tsionis, 2015). Numerous post-earthquake investigations have been





conducted for groups of buildings to derive the empirical damage matrices. A damage matrix is a table of predefined damage states and percentages of specific building types at which each damage state is exceeded due to particular macro-seismic intensity levels. However, as pointed out by Billah and Alam (2015), empirical investigations are usually limited to particular sites or seismo-tectonic/geotechnical conditions with abundant

seismic hazard and lack generality. Moreover, they usually refer to the macro-seismic intensity, which is not an instrumental measure but is based on a subjective evaluation (Maio and Tsionis, 2015). By contrast, analytical methods are based on static and dynamic nonlinear analyses of modelled buildings, which can produce slightly more detailed and relatively more transparent assessment algorithms with direct physical meaning (Calvi et al., 2006). Therefore, analytical methods are conceived to be more reliable than empirical results (Hariri-Ardebili

and Saouma, 2016). Nevertheless, variations in the different practices of analytical fragility studies, such as selection of seismic demand inputs, use of analysis techniques, characterisation of modelling structures, definition of damage states thresholds as well as usage of damage indicators by different authorities, can create discrepancies among various analytical results even for exactly the same building typology. In addition, analytical fragility studies for groups of buildings are computationally demanding and often technically difficult

to perform.

Despite the limitations of each fragility analysis method, both empirical and analytical fragility curves are essential in conducting seismic risk assessment. However, the application of the existing fragility curves has been considered as a challenging task, since different approaches and methodologies are spread across scientific journals, conference proceedings, technical reports and software manuals, hindering the creation of an integrated

framework that could allow the visualization, acquisition and comparison between all the existing curves (Maio and Tsionis, 2015). In this regard, the first purpose of this study is to describe and examine available fragility curves, specially developed for Chinese buildings from 87 papers and theses using empirical and analytical methods. The median fragility functions from these previous research findings for the main building types in seismic prone areas in mainland China are then outlined and compared with international projects with similar

focus.

Furthermore, based on the empirical and analytical fragility database collected, the second purpose of this work is to propose a new approach in deriving intensity-PGA relation by using fragility as the bridge. The main concern behind this attempt is that intensity-PGA relation is quite essential in seismic hazard assessment, while traditional practices in deriving such a relation are generally region-dependent and have large scatter (Caprio et

al., 2015). Traditionally, intensity-PGA relations are developed using instrumental PGA records and empirical intensity observations within the same geographical range. In this work, we try to establish intensity-PGA relation using fragility as conversion media. Formally, this is achieved by the elimination of the fragility values from the fragility–intensity and from the fragility–PGA relation. Theoretically, reasonable results should emerge if the building types used in analytic fragility analyses and those investigated in the empirical field surveys are

close enough.

This study is organized as follows. In Section 1, the necessity of fragility database construction and the pros and cons of main fragility analysis methods are briefly introduced. In Section 2, a literature review of fragility studies in mainland China and related concepts is provided. Section 3 presents the discrete fragility database extracted from reviewed papers and theses. In Section 4, median empirical and analytical fragility curves are derived for

major building types in seismic prone areas in mainland China. Comparisons with international projects with





similar focuses are also conducted in this section. In Section 5, we introduce in detail our new approach in developing intensity-PGA relation by using fragility as bridge, which is quite comparable with relation developed by traditional practice. In Appendix and Code/Data availability, accesses to supplementary documents mentioned in the context are provided.

## 2  Review of building fragility studies in mainland China

### 2.1  Empirical method

As documented in Calvi et al. (2006), the first application of empirical method to investigate building fragility at large geographical scale was carried out in the early 1970s. In mainland China, since the 1975 Haicheng M7.5 earthquake, around 112 post-earthquake surveys have been conducted for M≥4.7 earthquakes (Ding, 2016). Currently, the main processes in post-earthquake field investigation and macro-seismic intensity determination in mainland China basically follow the workflow proposed by Hu (1988) based on the field work of Tonghai earthquake in the 1970s (Wang et al., 2007). In this workflow, the key concept of "average damage index" is introduced. That means, in each post-earthquake field survey unit (village/town/street), the number of different types of buildings in each damage state are firstly investigated; median damage index of five damage states D5, D4, D3, D2, D1 as defined in GB17742-2008 are used in later on calculation, namely 0.93, 0.70, 0.43, 0.20, 0.05 for these five damage states respectively. For each building type in each field survey unit, the corresponding average damage index is derived by summarizing the products of percentage of building in each damage state and its damage index. Generally, there should be one or two predefined reference building types, thus the average damage index of other surveyed building types can be further scaled to the damage index of the reference building type. In the end, the overall average damage index for each survey unit is calculated by summarizing the products of each building type's scaled damage index and that building type's weight in the survey unit. Once the average damage index in the survey unit is determined, the corresponding macro-seismic intensity can be directly derived from the predefined empirical relation between macro-seismic intensity and damage index of reference building type (GB17742-2008). In mainland China, currently three reference building types are used to determine macro-seismic intensity: (1) Type A: wood-structure, soil/stone/brick-made old building; (2) Type B: single- or multi-storey brick masonry without seismic resistance; (3) Type C: single- or multi-storey brick masonry sustaining shaking of intensity degree VII. A detailed building structural damage state description for judgement of macro-seismic intensity scale in China is given in Table B2 (a non-official translation of the latest version of China seismic intensity scale: GB17742-2008).

Given the importance of building fragility in seismic risk assessment and loss mitigation, in total we reviewed 87 existing fragility analyses from papers and theses for the main building typologies in seismic prone areas in mainland China. It's worth to note that, in Ding (2016), a very detailed collection of empirical fragility database was provided for 112 M≥4.7 events since the 1975 M7.5 Haicheng earthquake based on available post-earthquake surveys. However, due to the lack of building seismic resistance capability information in this database, it is not suitable for our later-on fragility analysis. Thus, we did not use this database and instead collected our own empirical fragility database from individual publications and M.S./Ph.D theses. In mainland China, the main building types of concern are masonry and reinforced concrete (RC) buildings (Sun and Chen, 2009), given the wide distribution of masonry in rural and township areas and the increasing popularity of RC buildings in urban areas. Historic earthquakes that caused serious building damage mainly occurred in seismic



prone provinces including **Sichuan** (Chen et al., 2017; Gao et al., 2010; He et al., 2002; Li et al., 2015; Li et al., 2013; Sun et al., 2013; Sun et al., 2014; Sun and Zhang, 2012; Ye et al., 2017; Yuan, 2008; Zhang et al., 2016), **Yunnan** (He et al., 2016; Ming et al., 2017; Piao, 2013; Shi et al., 2007; Wang et al., 2005; Yang et al., 2017; Zhou et al., 2007; Zhou et al., 2011), **Xinjiang** (Chang et al., 2012; Ge et al., 2014; Li et al., 2013; Meng et al., 2014; Song et al., 2001; Wen et al., 2017), **Qinghai** (Piao, 2013; Qiu and Gao, 2015), **Fujian** (Bie et al., 2010; Zhang et al., 2011; Zhou and Wang, 2015) and **other seismic active zones** (A, 2013; Chen, 2008; Chen et al., 1999; Cui and Zhai, 2010; Gan, 2009; Guo et al., 2011; Han et al., 2017; He and Kang, 1999; He and Fu, 2009; He et al., 2017; Hu et al., 2007; Li, 2014; Liu, 1986; Lv et al., 2017; Ma and Chang, 1999; Meng et al., 2012; Meng et al., 2013; Shi et al., 2013; Sun and Chen, 2009; Sun, 2016; Wang et al., 2011; Wang, 2007; Wei et al., 2008; Wu, 2015; Xia, 2009; Yang, 2014; Yin et al., 1990; Yin, 1996; Zhang and Sun, 2010; Zhang et al., 2017; Zhang et al., 2014; Zhou et al., 2013). The main outputs of these post-earthquake surveys are empirical damage probability matrices (DPMs), which can be used to derive the discrete conditional probability of exceeding predefined damage limit states under different macro-seismic intensity degrees. That is, for the DPMs, macro-seismic intensity degree is usually used as the ground motion indicator. In mainland China, detailed definition of each intensity degree is provided in Chinese Official Seismic Intensity Scale GB17742-2008 (see the non-official English translation in Appendix Table B2).

## 2.2 Analytical method

As summarized in Introduction section, the main drawback of empirical method lies in the subjectivity on allocating each building to a damage state and the lack of accuracy in the determination of the macro-seismic intensity affecting the region (Maio and Tsionis, 2015). Furthermore, the interdependency between macro-seismic intensity and damage as well as the limited or heterogeneous empirical data are commonly identified as the main difficulties to overcome in the calibration process of empirical approaches (Del Gaudio et al., 2015). By contrast, analytical methodologies produce more detailed and transparent algorithms with direct physical meaning, that not only allow detailed sensitivity studies to be undertaken, but also allow for the straightforward calibration of the various characteristics of the building stock and seismic hazard (Calvi et al., 2006). Different from the empirical fragility that is directly collected from post-earthquake survey, the derivation of analytical fragility curve is often based on nonlinear fine-element analysis. Popular analytical methods include push-over analysis (Freeman, 1998; Freeman, 2004), adaptive push-over method (Antoniou and Pinho, 2004), and incremental dynamic analysis (IDA) (Vamvatsikos and Cornell, 2002; Vamvatsikos and Fragiadakis, 2010). Within these approaches, most of the methodologies available in literature lie on two main and distinct procedures: the correlation between acceleration or displacement capacity curves and spectral response curves, as the well-known HAZUS or N2 methods (FEMA, 2003; Fajfar, 2000), and the correlation between capacity curves and acceleration time histories, as proposed in Rossetto and Elnashai (2003).

The major steps in using analytical methods to study building fragility include: the selection of seismic demand inputs, the construction of building models, the selection of damage indicator and the determination of damage limit state criteria (Dumova-Jovanoska, 2000). To combine empirical post-earthquake damage statistics from actual building groups with simulated/analytical damage statistics from modelled building types under consideration, we examined quite a few studies deriving analytical fragility curves for masonry and RC buildings in mainland China. The analysis techniques in these studies vary from static push-over analysis or adaptive push-


over method (Cui and Zhai, 2010; Liu, 2017), to dynamic history analysis or incremental dynamic analysis (Liu et al., 2010; Liu, 2014; Liu, 2014; Sun, 2016; Wang, 2013; Yang, 2015; Yu et al., 2017; Zeng, 2012; Zheng et al., 2015; Zhu, 2010) as well as based on necessary statistical assumptions (Fang, 2011; Gan, 2009; Guo et al., 2011; Hu et al., 2010; Zhang and Sun, 2010).

**2.3 Damage state definition**

As predefined, building fragility describes the exceedance probability of specific damage state given an ensemble of earthquake ground motion levels. To describe the susceptibility of building structure to certain ground motion level, four damage limit states are used to discriminate between different strengths of ground shaking: slight damage (LS1), moderate damage (LS2), serious damage (LS3) and collapse (LS4). These four

limit states divide the building into five structural damage states, namely negligible (D1), slight damage (D2), moderate damage (D3), serious damage (D4) and collapse (D5). The relation between limit states and structural damage states is illustrated by Fig. 1. Hereafter, fragility curves in this study specifically refer to the probability of exceeding four damage limit states (LS1, LS2, LS3, LS4) under different ground motion levels.

Standard definitions of building structural damage states have been issued in different countries and areas. In the

European Macro-seismic Scale 1998 (EMS1998) proposed by European Seismological Commission (ESC), five grades of structural damage are defined: negligible to slight damage (Grade 1), moderate damage (Grade 2), substantial to heavy damage (Grade 3), very heavy damage (Grade 4) and destruction (Grade 5). In the HAZUS99 Earthquake Model Technical Manual, developed by Department of Homeland Security, Federal Emergency Management Agency of the United States (FEMA) in 1999, generally four structural damage classes

are used for all building types: slight damage, moderate damage, extensive damage and complete damage. Other damage state classifications like MSK1969 proposed by Medvedev and Sponheuer (1969) and AIJ1995 (Nakamura, 1995) in Japan issued by Architectural Institute of Japan are summarized in Table 1. In mainland China, the latest standard GB17742-2008 was issued in 2008 by China Earthquake Administration (CEA), in which detailed damage to structural and non-structural components are defined for each damage state (Table 2).

In empirical method, fragility curve is derived from damage probability matrices (DPMs) based on post-earthquake field surveys. DPMs give the proportions of buildings in each structural damage state (D1, D2, D3, D4, D5), and they can be used to derive the probability of exceeding each damage limit state $P[LS_i]$ ($i$=1,2,3,4), as illustrated in Eq. (1):

$$P[LS_i] = 1 - P[D_i] \ (i = 1); \quad P[LS_i] = P[LS_{i-1}] - P[D_i] \ (i = 2 \dots N) \tag{1}$$

where $N$ refers to the total number of damage limit states (here $N$=4); for each building type, $P[D_i]$ refers to the proportion of building in each structural damage state $i$.

In analytical method, fragility curve is derived by Eq. (2), with the assumption that building response to seismic demand inputs follows the lognormal distribution:

$$P[LS|S_d] = \Phi \left[ \frac{1}{\beta_{LS}} \ln(\frac{S_d}{S_{C|LS}}) \right] \tag{2}$$





where $P[LS|S_d]$ is the probability of being in or exceeding damage limit state $LS$ due to ground motion indicator $S_d$ (e.g. the inter-storey displacement); $S_{C|LS}$ refers to the median value of damage state indicator at which the building reaches the threshold of the damage state $LS$; $\beta_{LS}$ represents integrated uncertainties from seismic demand input, building capacity and model uncertainty, generally within the range of 0.6-0.8; $\Phi[\ ]$ is the normal cumulative probability distribution.

## 3 Fragility database analysis

### 3.1 Building typology and seismic resistance level classification

During the past four decades, more than 2000 M≥4.7 earthquakes have occurred in mainland China and its neighbouring areas (Xu et al., 2014). Up to 2014, post-earthquake field surveys have been conducted for at least 112 damaging earthquakes that occurred in the densely populated areas in mainland China, since the 1975 M7.3 Haicheng earthquake (Ding, 2016). These damaging earthquakes mainly clustered in seismic prone provinces in southwestern China (e.g. Sichuan, Yunnan) and western China (e.g. Xinjiang Uygur, Tibet, Qinghai), as shown in Fig. 2. The main building types in these areas are featured by masonry, reinforced concrete (RC), brick-wood, soil, stone as well as chuandou-timber (a typical building type in mountainous area of Tibet, Qinghai and Sichuan). Due to the limitation in fragility data abundance, we mainly focus on studying the seismic fragility of the two most widely distributed building types: masonry and RC buildings (Sun and Chen, 2009). Masonry buildings are mainly composed of brick and concrete. RC buildings include building structures such as RC core wall, frame structure and frame-shear wall.

The seismic resistance level of masonry and RC buildings is further divided into two classes: level A and level B. The assignment of seismic resistance level in this study is mainly based on supplementary information given in each scrutinized literature, including building age, construction material, seismic resistance code at construction time, load-bearing structure etc. Given the changes in building quality and corresponding code standard over the past four decades in China, buildings constructed in different ages though with the same nominal resistance level of each period, are reassigned with different seismic resistance levels according to the latest standard. The referred grouping criteria is given in Table 3 (*more building classification details can be found from the online supplement*). Generally, "level A" includes buildings with seismic resistance level assigned as pre/low/moderate-code, and "level B" includes buildings assigned as high-code.

### 3.2 Outlier check

After grouping the empirical and analytical fragilities based on building type (masonry and RC) and seismic resistance level (A and B) in Sect. 3.1, the empirical fragility database based on macro-seismic intensity (Fig. 3) and analytical fragility database based on PGA (Fig. 4) for four damage limit states (LS1, LS2, LS3, LS4) are thus constructed (*data can be found from the online supplement*). The Y-axis "fragility" of Fig. 3 and Fig. 4 refers to the exceedance probability of each damage limit state at each ground motion level. As can be seen, the scatter of fragilities vary across building types and seismic resistance levels. For empirical fragilities, the scatter may relate to the uneven abundance of damage data for buildings investigated in post-earthquake field surveys, the subjective judgement of damage states as well as the rough division of building structure types. For analytical fragilities, the scatter may come from the difference in the selection of seismic demand inputs, the use



of analysis techniques, the detailing of the modelled building structure, the definition of damage state as well as the difference in damage indicators used by different researchers. Thus, before deriving consecutive building fragility curves from these discrete fragility data in Fig. 3 and Fig, 4, the outliers need to be firstly removed from these originally collected datasets.

To figure out the outliers in the originally collected fragility database, the box-plot check method was applied. For each building type (Masonry_A, Masonry_B, RC_A, RC_B) and in each damage limit state (LS1, LS2, LS3, LS4), the corresponding series of fragility data was sorted from the lowest to the highest value. Three quantiles ($Q_1$, $Q_2$, $Q_3$) were used to divide each fragility series into four equal-sized groups and they correspond to the 25%, 50% and 75% quantile value in each series. A discrete fragility value ($Q_i$) was assigned as an outlier if

$Q_i - Q_3 > 1.5 \times (Q_3 - Q_2)$ or $Q_1 - Q_i > 1.5 \times (Q_2 - Q_1)$. The box-plot check results are shown in Fig. 5 for empirical fragility data and in Fig. 6 for analytical fragility data.

## 4    Derivation of representative fragility curves

After removing outliers, details of the remaining fragility dataset (e.g., the number of data points, median and standard deviation of these data) for each damage state of each building type are plotted in Appendix Fig. A1-A4

and summarized in Appendix Table B1. It is worth to iterate that, as aforementioned in the Introduction section, the organization of this study is centred on two focuses. The first one is to construct a comprehensive fragility database for Chinese buildings from 87 papers and theses using empirical and analytical methods, which is one key component of seismic risk assessment. Based on the empirical and analytical fragility database collected, the second focus is to propose a new approach in deriving intensity-PGA relation by using fragility as the bridge. In

this regard, a representative fragility curve should be firstly derived for each damage state of each building type, and we refer to use the median fragility values to derive such a curve.

### 4.1    Median fragility curve

To derive a representative fragility curve for each damage limit state (LS1, LS2, LS3, LS4) of each building type (Masonry_A, Masonry_B, RC_A, RC_B) for further study (to derive intensity-PGA relation in Sect. 5), the

median values (50% quantile) of each fragility series in Fig. 5 and Fig. 6 are used. For consecutive median fragility curve derivation, cumulative normal distribution is assumed to fit the discrete median empirical fragilities and log-normal distributions is assumed to fit the discrete median analytical fragilities. For each damage limit state of each building type, the parameters $\mu_{LS}$ and $\sigma_{LS}$ in the consecutive fragility curve can be regressed following Eq. (3):

$$P(X|LS) = \Phi\left[\frac{X_{int} - \mu_{LS}}{\sigma_{LS}}\right] \ or \ P(X|LS) = \Phi\left[\frac{1}{\sigma_{LS}}\ln\left(\frac{X_{PGA}}{\mu_{LS}}\right)\right] \qquad\qquad (3)$$

where $P(X|LS)$ represents the exceedance probability of each damage limit state $LS$ given ground motion level $X$ ($X$ refers to $X_{int}$, namely macro-seismic intensity in terms of empirical fragility; and $X$ refers to $X_{PGA}$, namely PGA in terms of analytical fragility).

The median fragility curves derived for empirical data and for analytical data are plotted in Fig. 7 and Fig. 8,

respectively. As can be clearly seen from Fig. 7 and Fig. 8, there are two obvious trends: (1) for the same building type (masonry or RC), the higher the seismic resistance level (A<B), the lower the building fragility,





which applies for all damage limit states; (2) for the same seismic resistance level, RC building has lower fragility than masonry building, which also applies for all damage limit states. These two trends indicate the reliability of the newly collected fragility database, the reasonability of the criteria in grouping building types and seismic resistance levels, as well as the suitability of using median fragility values to develop representative

fragility curves for further analysis. However, some extra abnormality is also noteworthy, e.g. in the median fragility curve developed for LS4 of "RC_B" in Fig. 8, the probability to exceed LS4 damage limit state remains 0 even when PGA is higher than 0.8 g, which is obviously not the case in reality. Detailed source of such abnormality and its effect on the intensity-PGA relation to develop will be discussed in Sect. 5.3.

Mathematically, the goodness of fit of the consecutive median fragility curve from discrete median fragilities can

be measured by statistical indicator $R^2$ (Draper and Smith, 2014). Higher $R^2$ value indicates a better fit of the regressed fragility curve, since it is defined as the ratio between *SSR* and *SST*: *SSR* is the *sum of squares of the* ***regression*** $(SSR = \sum_{i=1}^{n}(\hat{y}_i - \bar{y}_i)^2)$, and SST is the *total sum of squares* $(SST = \sum_{i=1}^{n}(y_i - \bar{y}_i)^2)$; $y_i$ refers to the original discrete fragilities for each damage limit state, $\bar{y}_i$ refers to the mean fragility, $\hat{y}_i$ refers to the predicted fragility by the fitted fragility curve. As shown in Table 4, the $R^2$ values are generally above 0.95, which

indicates the normal or lognormal distribution assumption in Eq. (3) is very suitable to match the discrete fragility datasets. Noticeably, there are also three low $R^2$ values (≤0.8) in Table 4 for damage limit state LS1, LS2, LS3 of building type "RC_A", which may indicate the low quality (e.g. high scatter) of the originally collected fragility data. As can be cross-validated from Fig. 4 and even better Fig. 6, the analytical fragility data for "RC_A" are more scattered than for other building types. This thus directly leads to the low $R^2$ values in

fitting the median fragility curve for damage limit state LS1, LS2, LS3 of "RC_A".

### 4.2 Fragility curve comparison with international projects

Given the import role of fragility curve for future seismic risk assessment and loss estimation especially in seismic prone areas, we construct the empirical and analytical fragility database from 87 individual papers and theses. The median and standard deviation of the fragility series for each damage state of each building type are

plotted in Appendix Fig. A1-A4 and listed in Appendix Table B1. To derive new intensity-PGA relation by using fragility as the bridge, the median fragility curves for masonry and RC building types with seismic resistance levels A and B are regressed, as shown in Fig. 7 and Fig. 8. While, the robustness of these median fragility curves varies across damage states and building types, as will be discussed in detail in Sect. 5.3. Here, to better place such a review work, it is worth to compare it with international projects with similar focus.

Therefore, we also checked the fragility outputs from several international projects, including PERPETUATE, SYNER-G, PAGER, HAZUS and GEM. Based on the detailed analysis in Sect. 5.3, median fragility curves developed for building type "Masonry_A" are of relatively highest robustness. Thus, the comparison of our median fragility curves with these international projects (in terms of PGA) will be based on "Masonry_A" building type only. To avoid over complexity, the scatter attached to each fragility curve is not taken into

account in this comparison.

### 4.2.1 PERPETUATE project

The main goal of European PERPETUATE project was to develop European guidelines for evaluation and





mitigation of seismic risk to cultural heritage assets, applicable in the European and other Mediterranean countries. As written in their project homepage, the risk assessment of heritage buildings requires the assessment of both architectonic and artistic assets contained in them, which needs improvement in methods of analysis and assessment procedures rather than in intervention techniques for common buildings and infrastructures. Besides that, a verification approach in terms of displacement rather than in terms of strength is more reliable and effective for heritage building. However, the fragilities collected in this study are mainly for general masonry and RC buildings and mainly in terms of macro-seismic intensity or PGA, not displacement. Therefore, the fragility outputs from PERPETUATE project and our study are not so comparable.

### 4.2.2 SYNER-G project

European SYNER-G project focused on reviewing fragility for masonry and RC buildings in Europe, including the collection of existing fragility functions, building recategorization, and harmonization of intensity measure and limit states. In the final output of SYNER-G project, fragility curves were given in terms of PGA, with some of them converted from macro-seismic intensity or spectral acceleration (SA) (Crowley et al., 2011a; Crowley et al., 2011b; Kaynia et al., 2013; Pitilakis et al., 2014). Comparisons between mean fragility curves developed in SYNER-G project (Crowley et al., 2011b; refer to Table 6.3 and Table 6.5) with our median analytical fragility curves for "Masonry_A" building type are plotted in Fig. A5. It is noteworthy that, in SYNER-G outputs, the typically four damage limit states were further harmonized into two: yielding damage state (LS2) and collapse state (LS4).

As can be seen from Fig. A5, the fragility in SYNER-G project for typical European masonry buildings is much higher than "Masonry_A" in this study. The discrepancy of fragility for masonry between SYNER-G (Silva et al., 2014) and this study is potentially related to the following factors. Firstly, the difference in use of ground motion indicator (part of SYNER-G's PGA-related fragility curves are converted from intensity, SA related fragility curves; our analytical fragility curves are purely PGA-related) may play a role. Besides that, difference in building classification is difficult to accurately calibrate but certainly contributes to the final fragility discrepancy. Furthermore, the harmonization of damage limit states in SYNER-G project (from four to two) makes the fragility difference between these two damage states subtle as shown in Fig. A5, and also hinders the fragility comparison for each damage limit state accordingly. Lastly, in SYNER-G project, the mean fragility curves are provided; but in this study, median fragility curves are developed for nominally similar masonry buildings. In all, these differences combined together lead to the fragility discrepancy for general masonry buildings in our fragility curves and those in SYNER-G project.

### 4.2.3 PAGER project

The ongoing PAGER project (Prompt Assessment of Global Earthquakes for Response) of United States is an automated system specialized in estimating the seismic ground shaking distribution, the number of people and settlements exposed to that severe shaking, and the scale/range of possible fatalities and economic losses. For these purposes, vulnerability functions are used, which are different from the fragility functions we focus on in this study. The main difference lies in that, vulnerability functions can be derived either directly from historic damage information, or indirectly from fragility function combined with consequence functions, which describe



the probability of loss given a level of performance (e.g. collapse damage state). Thus, direct comparison between the outputs of PAGER project and our study is not available.

### 4.2.4 HAZUS project

The ongoing HAZUS project of United States, is developed and updated to provide local, state and regional officials with the tools necessary to plan and stimulate efforts to reduce risk from earthquakes and to prepare for emergency response and recovery from an earthquake (FEMA, 2003; FEMA, 2008). HAZUS offers a series of fragility curves for typical building types based on HAZUS building taxonomy. Here for comparison with our fragility curves for "Masonry_A", we extracted HAZUS fragility curves developed for two specific masonry building types: mid-rise reinforced masonry bearing walls with wood or metal deck diaphragms (RM1M) and high-rise reinforced masonry bearing walls with precast concrete diaphragms (RM2H) from HAZUS Earthquake Technical Manual (refer to Table 5.16a-d). These fragility curves are based on median spectral displacement of the damage state of interest and an assumed demand spectrum shape that relates spectral response to PGA. The reference spectrum used to derive HAZUS fragility curve represents ground shaking of a large-magnitude (i.e. M ≈7.0) western United States earthquake for soil sites (e.g. site class D) at site to source distances of 15km or greater.

Comparisons between the fragility curves of HAZUS for "RM1M, RM2H" and those developed in this study for "Masonry_A" are plotted in Fig. A6 (RM1M) and Fig. A7 (RM2H). In HAZUS four code levels (pre/low/moderate/high) are provided; but in this study, only two seismic resistance levels "A" and "B" are assigned to the analytical fragilities extracted from individual literature (*more details can be found from online supplement*). As can be seen from Fig. A6 and Fig. A7, the median fragility of our "Masonry_A" is basically between the median fragility of high-code and moderate-code of "RM1M" and "RM2H" building types, which indicates the compatibility between HAZAU and our median fragility curves for general masonry buildings, compared with the mean fragility curves in SYNER-G project.

### 4.2.5 GEM project

The ongoing Global Earthquake Modelling (GEM) project, is motivated to "serve the public good in a collaborative, credible, open and transparent manner, and to make risk assessment inclusive to create a holistic culture of awareness and resilience, bringing a once-scarce resource available to all sectors and beneficiaries". GEM project has integrated outputs from three other European projects: SHARE, SYNER-G and NERA. SHARE focuses on seismic hazard harmonization in Europe and covers all of Europe and the Maghreb countries. The hazard model is developed with the OpenQuake Engine. SYNER-G partners developed a unified methodology and tools for systemic vulnerability assessment in Europe. NERA focused on the creation of a European research infrastructure for risk assessment and mitigation. Besides the fragility outputs from SYNER-G project, GEM online fragility database also integrates fragility curves from HAZUS (Yepes-Estrada et al., 2016). Therefore, we do not repeat the comparison with GEM fragility curves developed for European and American building types. For mainland China, the fragility curves integrated in GEM database is solely from Tang et al. (2011) developed for Chinese schools, and only for RC building with spectral acceleration (SA) as the ground motion indicator. To avoid uncertainty introduced from converting SA to PGA, here we also skip the comparison with fragility curves in Tang et al. (2011).



## 5 New approach in deriving intensity-PGA relation

Intensity-PGA relation has an important application in seismic hazard assessment, since the use of macro-seismic data can compensate for the lack of ground motion records and thus help in reconstructing the shaking distribution for historical events. Traditionally, intensity-PGA relations are developed using instrumental PGA records and macro-seismic intensity observations within the same geographical range (Bilal and Askan, 2014; Caprio et al., 2015; Ding et al., 2014; Ding, 2016; Ding et al., 2017; Ogweno and Cramer, 2017; Worden et al., 2012). These relations are generally region-dependent and have large scatter (Caprio et al., 2015). In this section, we propose a new approach in deriving intensity-PGA relation based on the newly collected empirical and analytical fragility database. For each building type and each damage limit state, an empirical fragility curve (exceedance probability vs. macro-seismic intensity) and an analytic fragility curve (exceedance probability vs. PGA) are available, as derived from the median fragilities in Sect. 4.1. By eliminating the same fragility value, we can derive the corresponding pair of macro-seismic intensity and PGA. Thus, for a series of fragility values, we can further regress the corresponding intensity-PGA relation based on the paired intensities and PGAs. Ideally, we would expect the overlap of all these regressed intensity-PGA relations, regardless of the difference in building type, seismic resistance level and damage state.

### 5.1 Difference between this new approach and previous practices

Compared with this new approach in intensity-PGA relation development, previous practices directly regressed intensity and PGA datasets within the same geographical range, but no further classification of datasets was conducted, as based on building type or damage state in this study. The lack of further classification of PGA and intensity datasets may explain why the previously derived intensity-PGA relations generally have high scatter. The reason lies behind is that, although macro-seismic intensity is a direct macro indicator of building damage, higher instrumental ground motion (e.g., PGA) does not necessarily mean higher damage to all buildings. Instead, damage is more determined by the seismic resistance capacity of different building types. Thus, further division of intensity and instrumental ground motion records based on affected building types should promisingly help decrease the scatter of regressed intensity-PGA relation.

Furthermore, local site effect also contributes to the amplification of instrumental peak ground motions (PGA or SA), when combining intensity and PGA datasets from areas with different geological background together. This in turn increases the scatter of regressed intensity-PGA relation. In this regard, it is worth to emphasize that, in our PGA-related analytical fragility database, the PGA parameter is not the real instrumental records as used in regressing traditional intensity-PGA relation, but the input PGA records used in experimental fragility analysis (push-over analysis, incremental dynamic analysis, dynamic history analysis etc.). Therefore, the regional dependence (here we mainly refer to site condition), which contributes to the scatter of traditional PGA-intensity relation, is not a source of uncertainty in our relation.

### 5.2 Derivation of initial intensity-PGA relation

As a tentative approach, here we derive relation between intensity and PGA using median fragility as the bridge for each damage limit state of each building type. We're deeply aware that uncertainty is inherent in every single step both in empirical and analytical fragility analysis. However, the trial of using the median fragility as the





bridge to develop intensity-PGA relation, more importantly, aims at providing a new approach in this regard compared with traditional practice, not to backwards reduce the uncertainties (building structure, seismic demand inputs, computation methods etc.) in deriving empirical and analytical fragility. By using Eq. (3) for PGA-fragility and intensity-fragility respectively and eliminating fragility as variable, we find:

$\ln(PGA) = \alpha + \beta * Int,$

$with\ \alpha = \ln(\mu_{PGA}) - \frac{\sigma_{PGA}}{\sigma_{Int}} * \mu_{Int}, \beta = \frac{\sigma_{PGA}}{\sigma_{Int}}$ (4)

In which, the parameters $\mu_{PGA}, \mu_{Int}, \sigma_{PGA}, \sigma_{Int}$ are taken from Table 4 with values varying across building types and damage limit states.

These intensity-PGA relations are plotted in Fig. 9 (grouped by building types) and Fig. 10 (grouped by damage limit states). Theoretically, higher damage states can occur only for higher intensities or PGA values. For instance, a LS4 damage state at intensity III would not happen, as reflected by the curves in Fig. 9 and Fig. 10: LS1 have the lowest PGA or intensity starting point, while LS4 has the highest. Thus, we plot the intensity-PGA curves for fragility values above 1%. Ideally, we would expect the overlap of all relation curves between intensity and PGA, whether grouped by building type or by damage state. As a matter of fact, for building type "Masonry_A" and "Masonry_B" in Fig. 9, the four intensity-PGA curves of four damage limit states coincide very well. Meanwhile, the discrepancy in intensity-PGA relations of "RC_A" for damage states LS1, LS2, LS3 in Fig. 9 is not surprising, given the relatively high scatter in the original analytical fragility datasets of "RC_A" (as discussed in Sect. 4.1 and verified by Appendix Fig. A1-A4).

**5.3 Source of abnormality in intensity-PGA curves**

For building type "RC_A" and "RC_B" in Fig. 9, it is clear that for the same intensity levels, the corresponding PGA values of damage state LS4 are much higher than that of damage limit states LS1, LS2, LS3. For fixed fragility value, this may due to the underestimation of intensity by the median empirical fragility curve in Fig. 7, or the overestimation of PGA by the median analytical fragility curve in Fig. 8, or a combination of both effects. In this regard, damage data scarcity at higher damage limit states may contribute to the abnormal high PGA values of LS4. When reviewing the fragility data collection process, it is clear that the construction of empirical fragility database requires the combination of damage statistics from multiple earthquake events that cover a wide range of ground motion levels. Generally, large magnitude earthquakes occur more infrequently in densely populated areas, thus damage data tend to cluster around the low damage states and ground motion levels. This limits the validation of high damage states or ground motion levels (Calvi, 2006). According to Yuan (2008), those seriously damaged buildings in earthquake affected area are mainly masonry buildings. Therefore, the cause of the abnormal high PGA values of damage state LS4 for "RC_A" and "RC_B" can be attributed to the relative scarcity of damage data at higher intensity/PGA level, especially for RC buildings.

As to building type "masonry _A" and "masonry _B" in Fig. 9, for the same intensity level, the PGA values revealed by four damage states of "masonry _B" are generally higher than that in "Masonry_A". This can be more clearly seen from Fig. 10, in which the intensity-PGA relations are grouped by damage limit states. How to explain this abnormal phenomenon that given the same intensity level, the PGA values revealed by "Masonry_B" are generally higher than by all the other three building types? In fact, compared with "Masonry_A", buildings


assigned as type "Masonry_B" generally have much higher seismic resistance capacity. In this study, level "A" refers to buildings assigned as pre/low/moderate-code seismic resistance capacity, and level "B" refers to buildings assigned as high-code (*building classification details can be found on the online supplementary material*). That is, according to the grouping criteria in Table 3, buildings assigned as "Masonry_B" mainly refer to those built after 2001 with seismic resistance level VIII and above. This is obviously a very high code standard. Thus, for the same ground motion level, the damage posed on "Masonry_B" should be much slighter than on "Masonry_A". Therefore, for the same PGA level, the corresponding intensity revealed by "mansory_B" should be lower than by "mansory_A".

Actually in mainland China, the macro-seismic intensity level in post-earthquake filed surveys is determined by damage states of three reference buildings types, namely (1) Type A: wood-structure, soil/stone/brick-made old building; (2) Type B: single- or multi- storey brick masonry without seismic resistance; (3) Type C: single- or multi- storey brick masonry sustaining shaking of intensity degree VII. While in this study, buildings assigned as "Masonry_B" mainly refer to those constructed after 2001 with seismic resistance level VIII and above, and their seismic resistance capability is obviously much higher than all those three referred Type A/B/C building types. Therefore, intensity levels derived from damage to those less fragile "Masonry_B" buildings tend to be underdetermined. This may help explain why for the same intensity level, the corresponding PGA revealed by intensity-PGA relation of "mansory_B" is higher than that of "mansory_A".

Based on above discussion and the initial analysis in Sect. 4.1, it is clear that (a) Due to the high scatter in originally collected fragility database (as validated by the low $R^2$ values in Table 4), the intensity-PGA relations derived for LS1, LS2, LS3 are of low robustness; (b) Due to the damage data scarcity, intensity-PGA relations for LS4 of "RC_A" and LS4 of "RC_B" are also not fully reliable; (c) Due to the high seismic resistance capability attached to "Masonry_B", the intensity-PGA relations derived for all damage limit states of "Masonry_B" have the probability to underestimate intensity (or overestimate PGA) compared with "Masonry_A". Therefore, intensity-PGA curves derived for "Masonry_A" are of relatively highest robustness/reliability. Actually, the four intensity-PGA curves of "Masonry_A" do coincide very well as expected (Fig. 9). According to Yuan (2008), those seriously damaged buildings in earthquake affected areas are mainly masonry buildings. Therefore, we consider the median empirical and analytical fragility curves derived for "Masonry_A" (with uncertainties shown in Appendix Fig. A1-A4 and Table B1) are also the most representative ones for seismic prone areas in mainland China, compared with those developed for other buildings types in this study.

**5.4 Average intensity-PGA relation derived for "Masonry_A"**

According to the analysis in Sect. 5.3, intensity-PGA curves derived for four damage limit states of "Masonry_A" are of relatively highest robustness and have no such intensity underestimation probability as "Masonry_B". Therefore, we first focus only on building type "Masonry_A" and average its four curves for discrete intensity values, to derive the corresponding averaged PGA values, as listed in Table 5. If we match the data points in Table 5 with a linear relation between intensity and ln(PGA), we find Eq. (5):

$$\ln(PGA) = 0.521 * Int - 5.43 \pm \varepsilon \qquad (PGA: g) \qquad (5)$$

where ε follows the normal distribution, with 0 as the median value and the standard deviation is σ.



By integrating the uncertainty in both original empirical and analytical fragility data of "Masonry_A" (as shown in Appendix Fig. A1-A4 and Table B1) into the intensity-PGA relation, the averaged standard deviation σ in Eq. (5) is estimated to be 0.3 (*the detailed uncertainty transmission methodology is given in Appendix C*). As the "Masonry_A" type is the most common and relevant with buildings damaged in historic earthquakes (Sect. 5.3), we recommend using Eq. (5) for building damage assessment for earthquakes occurred in mainland China, especially in seismic active provinces e.g. Sichuan and Yunnan (Fig. 2).

### 5.5 Comparison with other intensity-PGA relations

Based on the analysis in Sect. 5.3, if we only remove those obviously unreliable intensity-PGA curves, namely LS1, LS2, LS3, LS4 of "RC_A" and LS4 of "RC_B", the range of PGA values derived from the remaining intensity-PGA relations are shown in Table 6. For comparison, the recommended PGA range for each intensity degree in the Chinese Official Seismic Intensity Scale (GB17742-2008) is listed in Table 7. The PGA values for intensity VI, VII in our results are higher than those in GB17742-2008; while for intensity VII, IX and X, the PGA values are quite comparable. We also found that the recommended PGA ranges in GB17742-2008 are indeed the same as those given in GB17742-1980, which was issued in the 1980s around four decades ago. At that time, available damage data used to derive the intensity-PGA relation in China was quite scarce. Therefore, damaging earthquakes occurred in the United States before 1971 were also largely used, which may not be representative of the situation in China today. Thus, one possible explanation for the relatively low PGAs for low intensity levels (VI, VII) in Table 7 (GB17742-1980/2008) is that, the buildings in the 1980s were more fragile than nowadays buildings. Since macro-seismic intensity is a direct macro indicator of building damage, nowadays buildings generally have better seismic resistance capacity and thus require higher ground motion (PGA) than buildings in the 1980s to be equally damaged.

Since the recommended PGA ranges in GB17742-2008 are not so representative of the current building status in mainland China, comparisons with the latest intensity-PGA relation developed in Ding et al. (2017) are also conducted. Ding et al. (2017) adopted traditional practice in regressing the macro-seismic intensities and instrumental PGA records within the same geographical range, by using records for 28 M≥5 earthquakes occurred during 1994-2014 in mainland China. The PGA values for intensity VI-IX in Ding et al. (2017) are listed in Table 8. When comparing our results in Table 5 and Table 6 with that in Table 8, PGA values are quite consistent for both low intensity (VI, VII) and high intensity (VIII, IX) levels, though these data are separately developed by our new approach and by traditional practice. This congruence shows the reasonability of our new approach proposed here in developing intensity-PGA relation.

### 6    Conclusion

We established empirical fragility database by evaluating 69 papers and theses, mostly from the Chinese literature, that document observations of macro-seismic intensities reflecting earthquake damage occurred in densely populated areas in mainland China over the past four decades. These publications provide empirical fragilities dependent on macro-seismic intensities for four damage limit states (LS1, LS2, LS3, LS4) of four building types (Masonry_A, Masonry_B, RC_A, RC_B). We also established analytical fragility database by scrutinizing 18 papers and theses with results on modeling fragilities for the nominally same building types and



the same damage states either by response spectral methods or by time-history response analysis. These analytic methods provide fragilities as functions of PGA. From this wealth of data, we derived the median fragility curves for these building types by removing outliers. Based on the newly collected empirical and analytical fragility database, possible comparisons with several international projects including PERPETUATE, SYNER-G, PAGER, HAZUS, GEM were also conducted.

We proposed a new approach by using fragility as the bridge and derived intensity-PGA relations independently for each building type and each damage state. The potential sources of abnormalities in these newly derived intensity-PGA relations were discussed in detail. Ideally the individual intensity-PGA curves should all coincide and allow us to derive an average relation between intensity and PGA. The coincidence is not 100% perfect and deviations for the cases where they occurr were discussed. Given the high damage data abundance and wide distribution of masonry buildings in mainland China, for studies referring to historic earthquakes and their losses in seismic active regions, e.g. Sichuan and Yunnan, we recommend utilizing the intensity-PGA relation derived from "Masonry_A" buildings in Eq. (5).

However, for engineering application, due to the scattering in original fragility datasets and simplification in using median fragility to derive intensity-PGA relation in our proposed new approach, the use of the preliminary intensity-PGA relations developed here should be with caution. It's also worth to note that, buildings used for empirical intensity determination and for analytical studies do not coincide: a "Masonry_A" building in a post-event field survey may encompass a wider range than in an analytic study. Therefore, following the novel idea of using fragility as the bridge to develop intensity-PGA relation in this study, possible extensions in the future can be performing fragility analysis for more specifically designed building types that are more representative of those widely damaged building types in the fields.

**Appendix**

To illustrate the scatter of the original fragility datasets we collected, error-bar analysis was performed and shown in Fig. A1 (empirical data) and Fig. A2 (analytical data) for all building types and seismic resistance levels. Specifically, to better scale the scatter, standard deviations of fragility, namely the exceedance probability of each damage limit state are provided in Fig. A3 (empirical data) and Fig. A4 (analytical data), respectively. Comparisons with fragility curves extracted from international projects including SYNER-G and HAZUS for masonry buildings are plotted in Fig. A5-A7.

In Table B1, more statistical details about our newly constructed fragility datasets, including the number of fragility data before and after removing the outliers, median fragility values used in deriving fragility curve in Fig. 7 and Fig. 8 and the standard deviation of each fragility dataset for each building type and each damage state are listed. Table B2 provides a non-official English translation of China seismic intensity scale: GB17742-2008.

Appendix C provides the methodology in transmission of uncertainty from empirical/analytical fragility database to intensity-PGA relation in Eq. (5).



**Code/Data availability**

More fragility extraction and building classification details are available from online supplement in:
(Filename: *Supplementary_building_classification_details.pdf*).
The earthquake catalog in plotting Fig. 2 is in:
5  (Filename: *EQ_list_with_field_survey.xlsx*).
The empirical and analytical fragility data in Fig. 3 and Fig. 4 are available in:
(Folder name: *data_Fig3-4*).

**Author contribution**

JD proposed the idea to review the fragility literature for buildings in mainland China. DX conducted the review
10  work and proposed the new approach in deriving intensity-PGA relation and wrote the text content. FW
proposed the methodology of uncertainty transmission from fragility to intensity-PGA relation. All authors
joined the revision of the manuscript.

**Competing interests**

The authors declare that they have no conflict of interests.



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





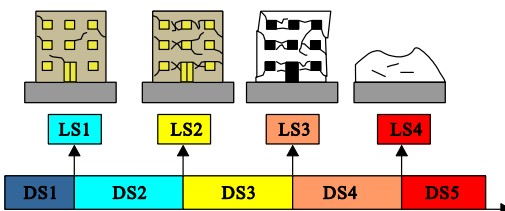

**Figure 1:** Corresponding Relation between structural damage states (DS1, D2, D3, DS4, DS5) and limit states (LS1, LS2, LS3, LS4) (modified from Wenliuhan et al., 2015).

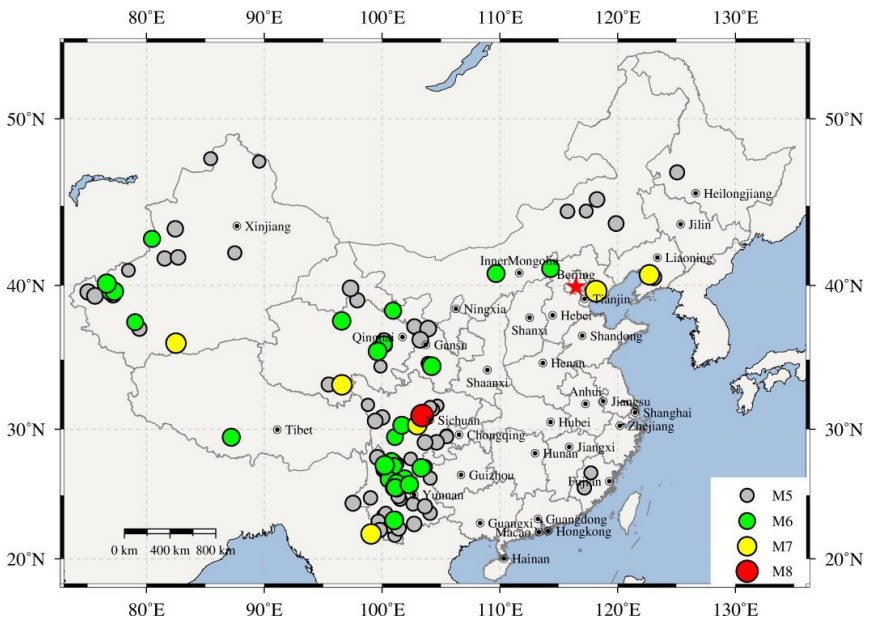

**Figure 2:** The distribution of earthquakes occurred in mainland China and its neighbouring area, for which field surveys were conducted. Detailed earthquake catalogue can be found from the online supplement, which is newly compiled based on Ding (2016) and Xu et al. (2014).

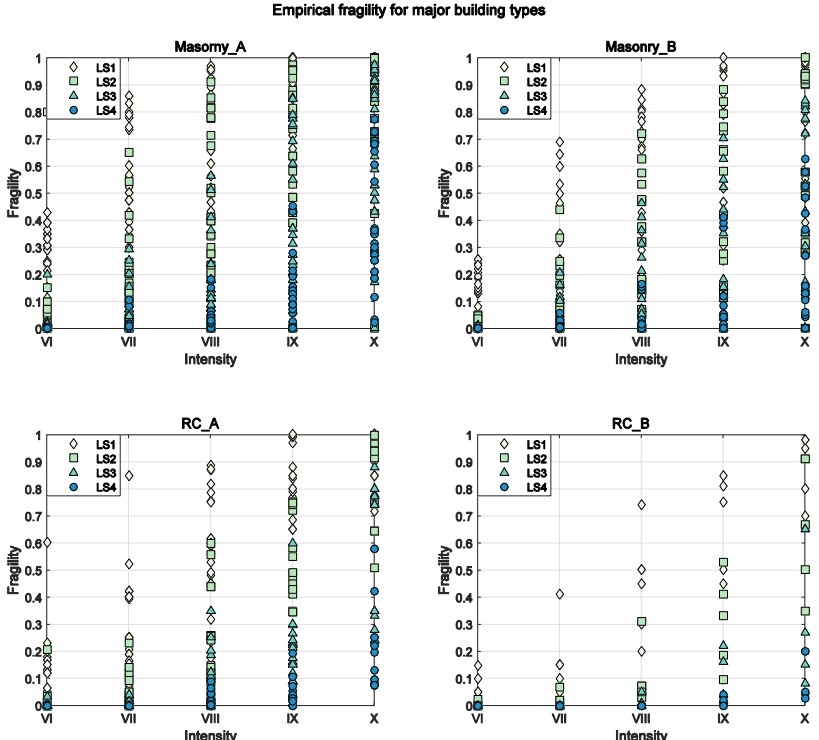

**Figure 3:** The distribution of empirical fragility data from post-earthquake field surveys, depicting the relation between the exceedance probability of each damage limit state (LS1, LS2, LS3, LS4) at given macro-seismic intensity levels. The fragility datasets are grouped by building types (masonry and RC) and seismic resistance levels (A and B).


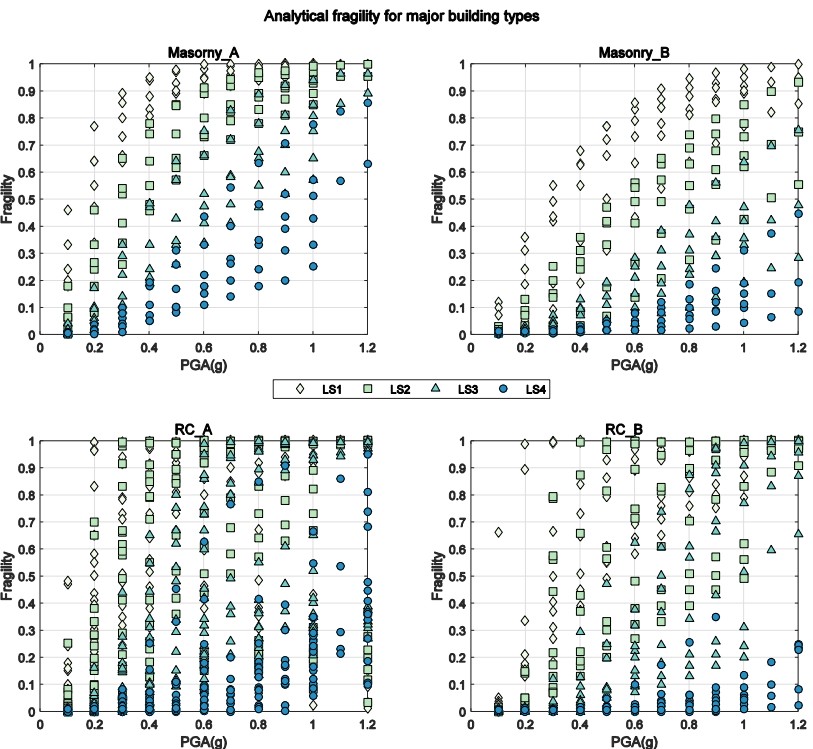

**Figure 4:** The distribution of analytical fragility data derived from non-linear analyses, depicting the relation between the exceedance probability of each damage limit state (LS1, LS2, LS3, LS4) at given PGA levels. The fragility datasets are grouped by building types (masonry and RC) and seismic resistance levels (A and B).

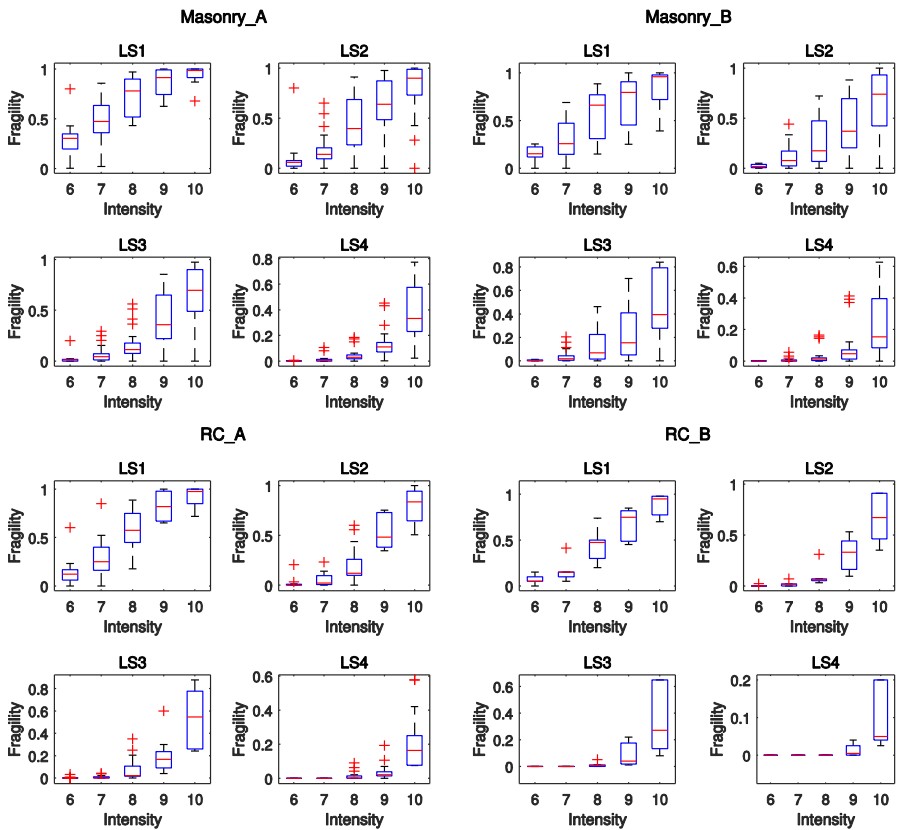

**Figure 5:** Outlier-check using box-plot method for empirical fragility data. Five macro-seismic intensity levels are used to classify the original fragility datasets: VI, VII, VIII, IX, X. "A" and "B" represent the pre/low/moderate-code and high-code seismic resistance level, respectively (more classification details are available from online supplement). LS1, LS2, LS3, LS4 are the four damage limit states. Outliers are marked by red crosses and red lines within each box indicates the 50% quantile fragility value.

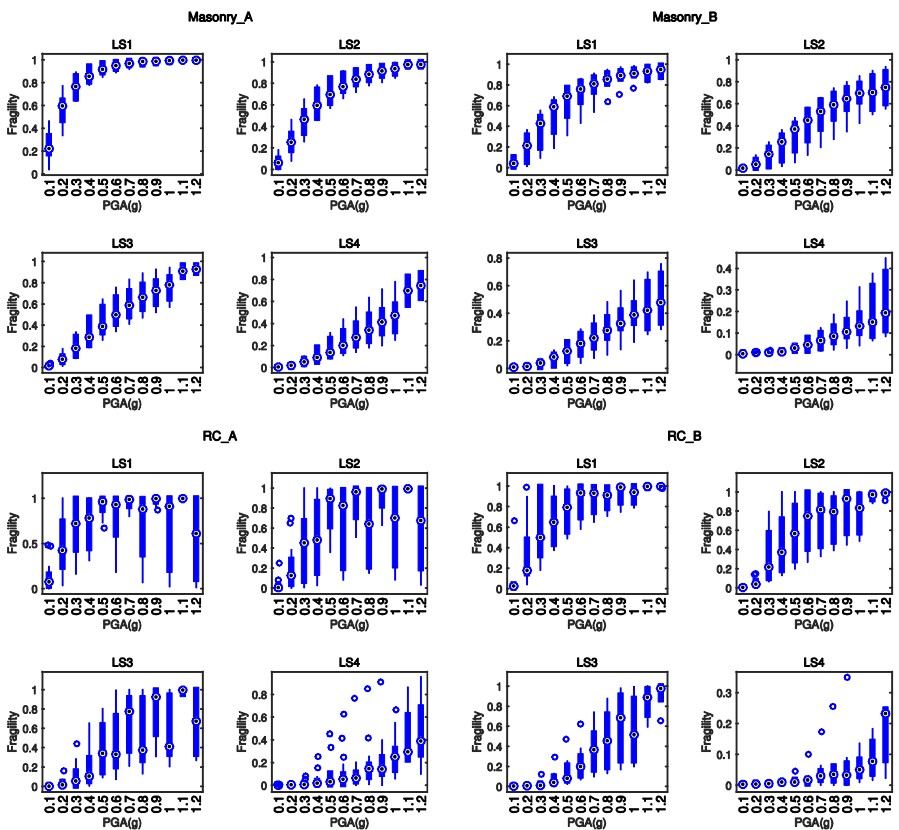

**Figure 6:** Outlier-check using box-plot method for analytical fragility data. Twelve PGA levels are used to group the discrete analytical fragility datasets: 0.1-1.2 g. "A" and "B" represent the pre/low/moderate-code and high-code seismic resistance level, respectively (more classification details are available from online supplement). LS1, LS2, LS3, LS4 are the four damage limit states. Outliers are marked by blue hollow circles and blue dot within each box indicates the 50% quantile fragility value.
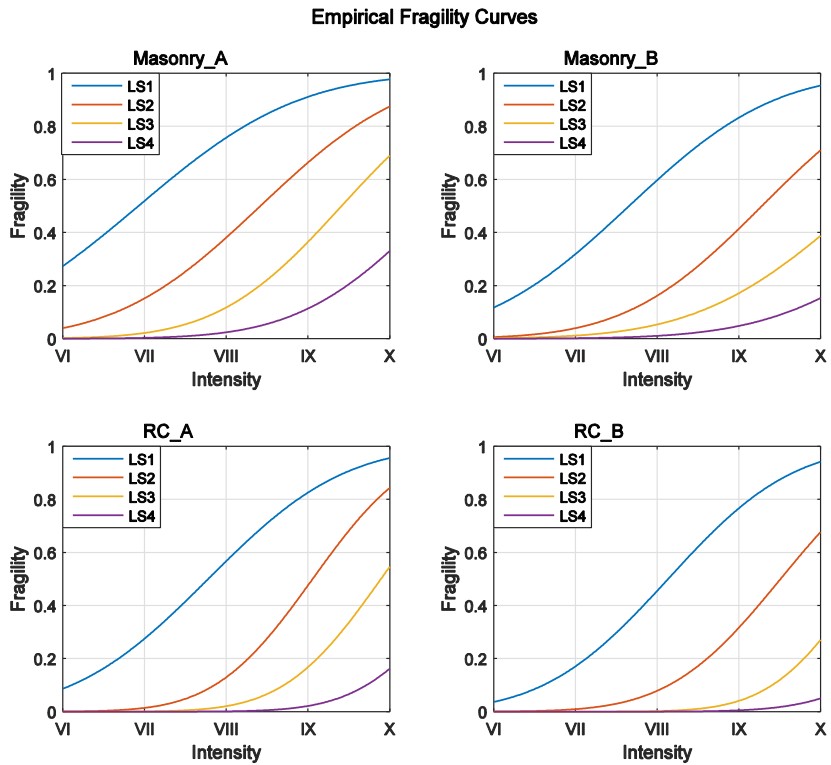

**Figure 7:** Median fragility curves derived from empirical fragility datasets, which depict the relation between macro-seismic intensity and exceedance probability of each damage limit state (LS1, LS2, LS3, LS4) for masonry and RC building types (Note: these median fragility curves are of varying robustness; see Sect. 4.1 and Sect. 5.3 for more details).

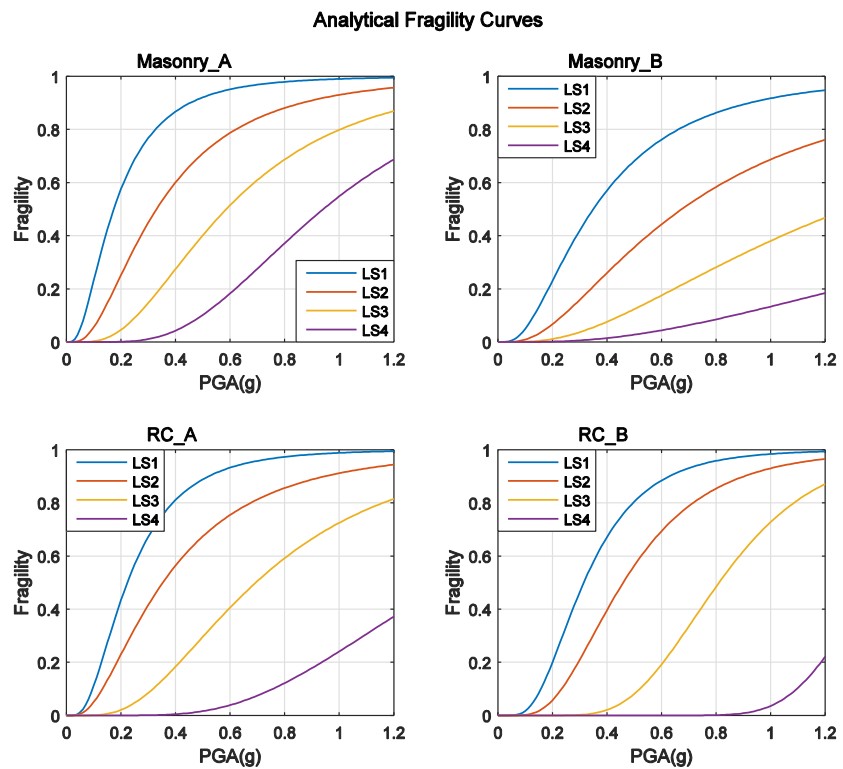

**Figure 8:** Fragility curves derived from analytical fragility datasets, which depict the relation between PGAs (unit: g) and exceedance probability of each damage limit state (LS1, LS2, LS3, LS4) for masonry and RC building types (Note: these median fragility curves are of varying robustness; see Sect. 4.1 and Sect. 5.3 for more details).

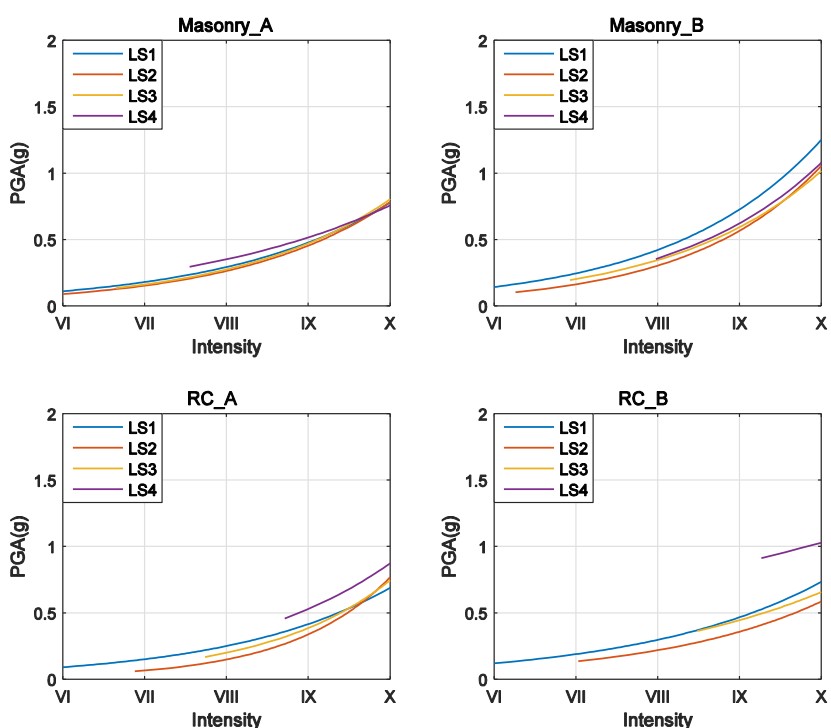

**Figure 9:** Intensity-PGA relations grouped by building types. Only intensity and PGA values with truncated exceedance probability ≥1% for each damage limit state of each building type are plotted, since higher damage states can appear only for higher intensities or PGA values (see Sect. 5.2 for more details).

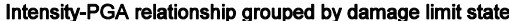

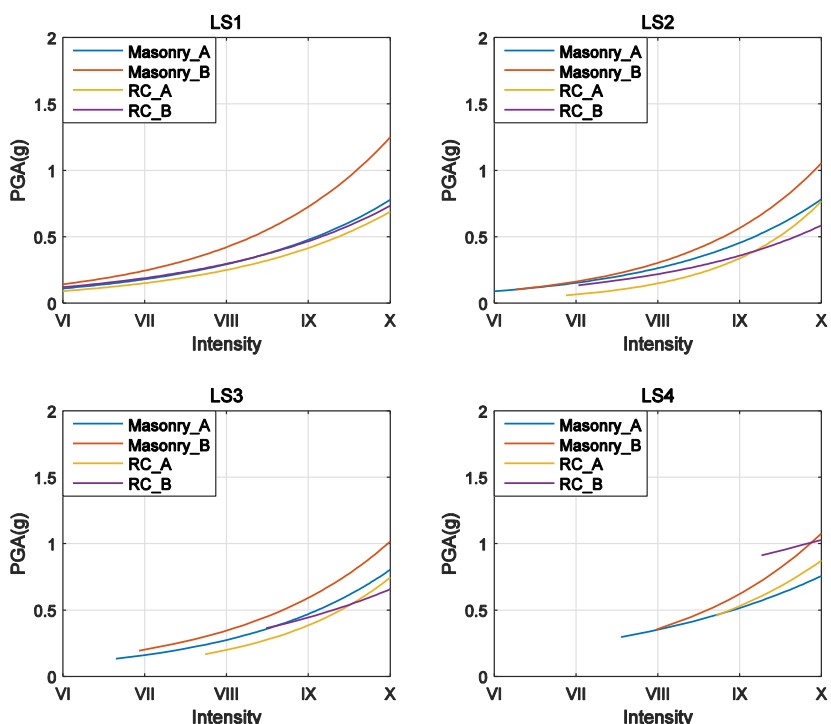

**Figure 10:** Intensity-PGA relations grouped by damage limit states. Only intensity and PGA values with truncated exceedance probability ≥1% for each damage limit state of each building type are plotted, since higher damage states can appear only for higher intensities or PGA values (see Sect. 5.2 for more details).

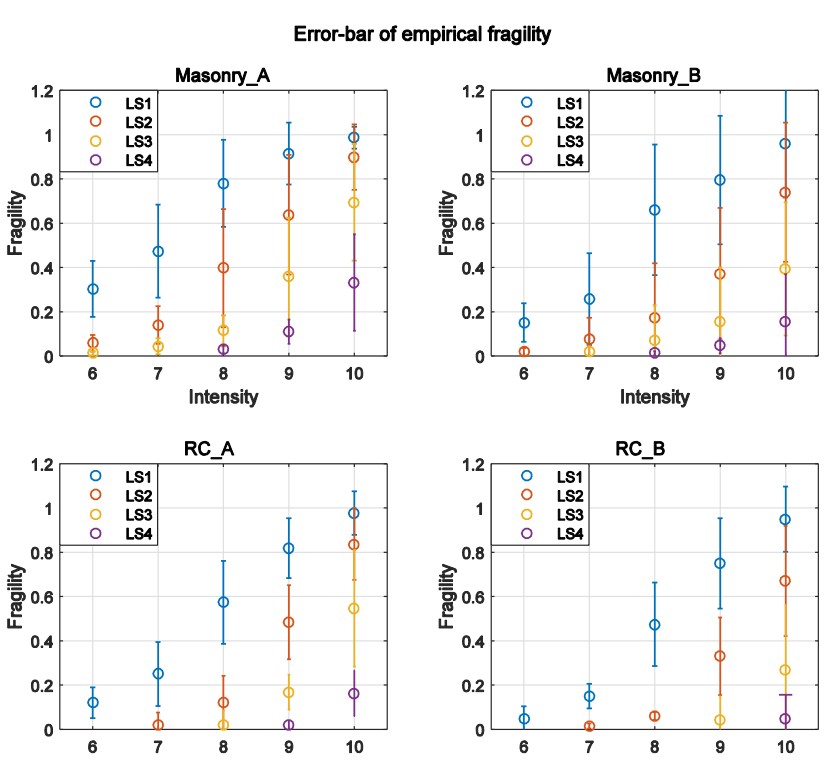

**Figure A1:** The error-bar of empirical fragility, namely the exceedance probability of each damage limit state (LS1, LS2, LS3, LS4) derived from empirical fragility datasets for each building type (Masonry_A, Masonry_B, RC_A, RC_B). Detailed values are given in Table B1. The circle within each bar represents the median exceedance probability of each damage limit state; the length of each bar indicates the value of the corresponding standard deviation. Only intensity and PGA values with truncated exceedance probability ≥1% for each damage limit state of each building type are plotted, since higher damage states can appear only for higher intensities or PGA values (see Sect. 5.2 for more details).

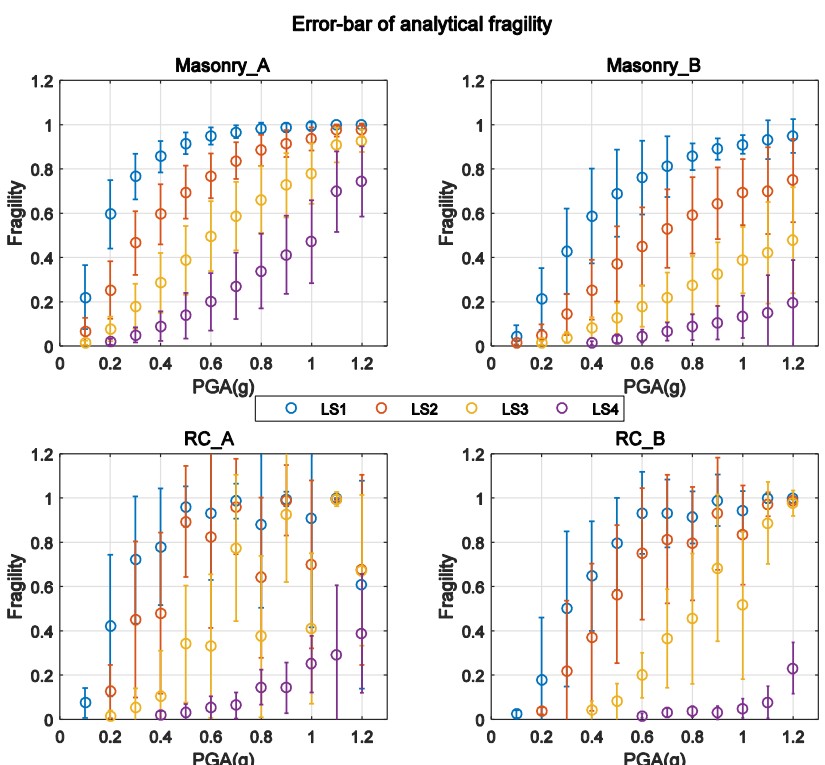

**Figure A2:** The error-bar of analytical fragility, namely the exceedance probability of each damage limit state (LS1, LS2, LS3, LS4) derived from analytical fragility datasets for each building type (Masonry_A, Masonry_B, RC_A, RC_B). Detailed values are given in Table B1. The circle within each bar represents the median exceedance probability of each damage limit state; the length of each bar indicates the value of the corresponding standard deviation. Only intensity and PGA values with truncated exceedance probability ≥1% for each damage limit state of each building type are plotted, since higher damage states can appear only for higher intensities or PGA values (see Sect. 5.2 for more details).

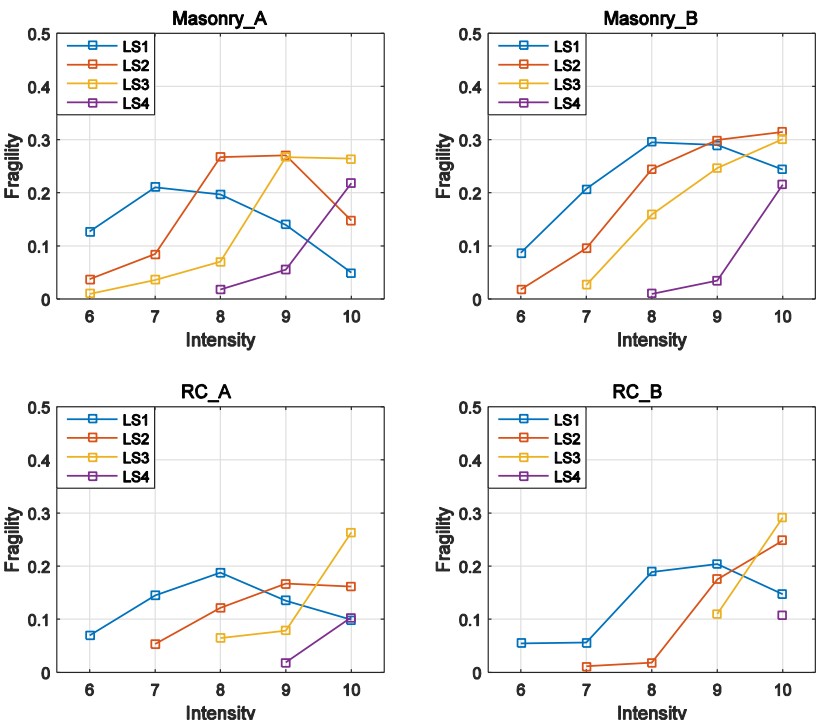

**Figure A3:** Standard deviation of empirical fragility, namely the exceedance probability of each damage limit state (LS1, LS2, LS3, LS4) derived based on empirical fragility datasets for each building type (Masonry_A, Masonry_B, RC_A, RC_B; detailed values are given in Table B1). Only intensity and PGA values with truncated exceedance probability ≥1% for each damage limit state of each building type are plotted, since higher damage states can appear only for higher intensities or PGA values (see Sect. 5.2 for more details).
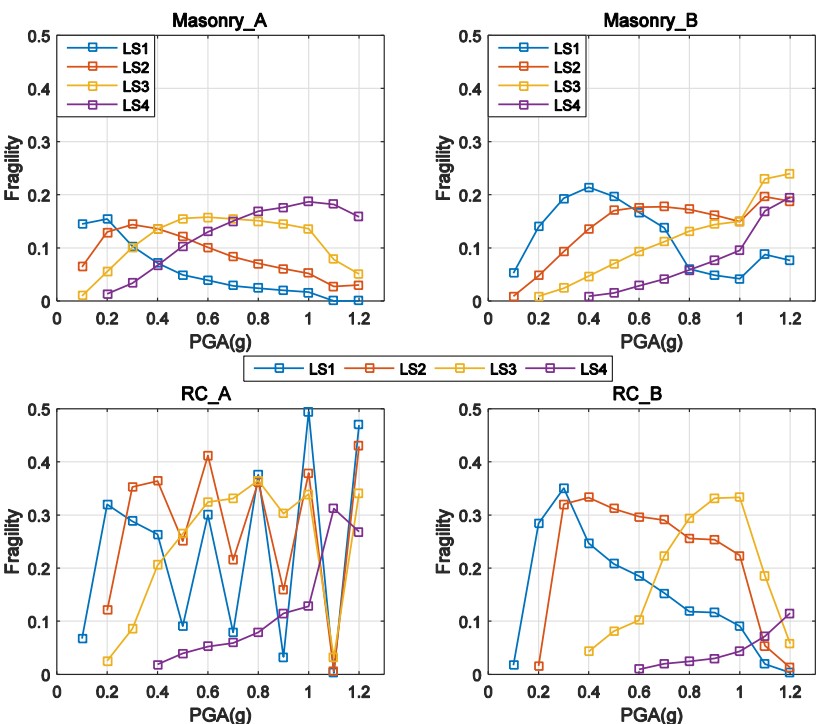

**Figure A4:** Standard deviation of analytical fragility, namely the exceedance probability of each damage limit state (LS1, LS2, LS3, LS4) derived based on analytical fragility datasets for each building type (Masonry_A, Masonry_B, RC_A, RC_B; detailed values are given in Table B1). Only intensity and PGA values with truncated exceedance probability ≥1% for each damage limit state of each building type are plotted, since higher damage states can appear only for higher intensities or PGA values (see Sect. 5.2 for more details).


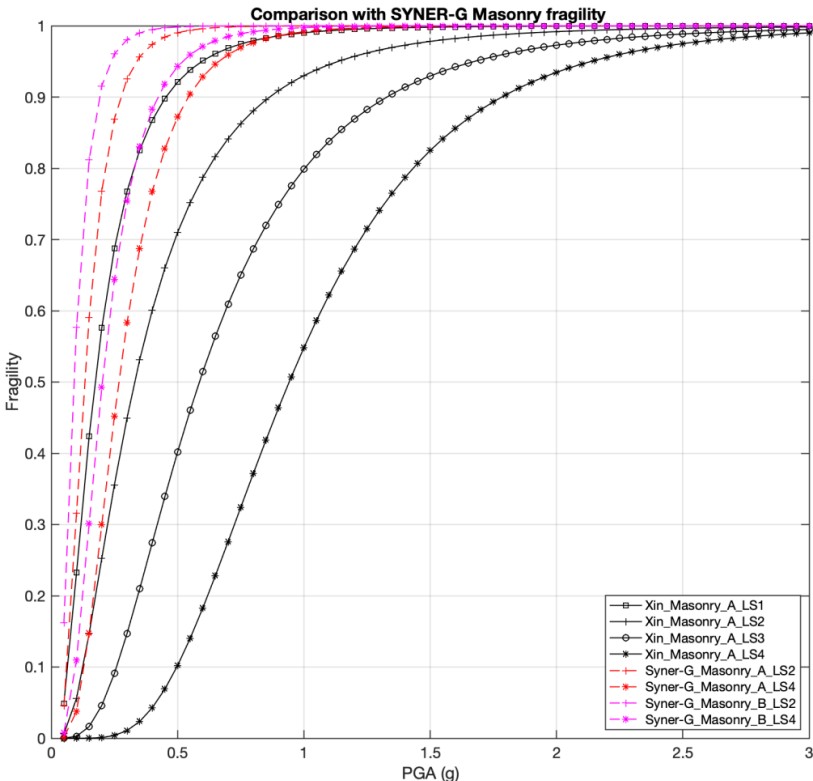

**Figure A5:** Fragility curve comparison between SYNER-G project mean outputs and our median results for masonry building type. In SYNER-G project, "Masonry_A", "Masonry_B" refer to the low-rise, mid-rise masonry building types, respectively; LS2 and LS4 refer to yielding state and collapse state (see Sect. 4.2.2 for more detailed discussion on sources of discrepancy).
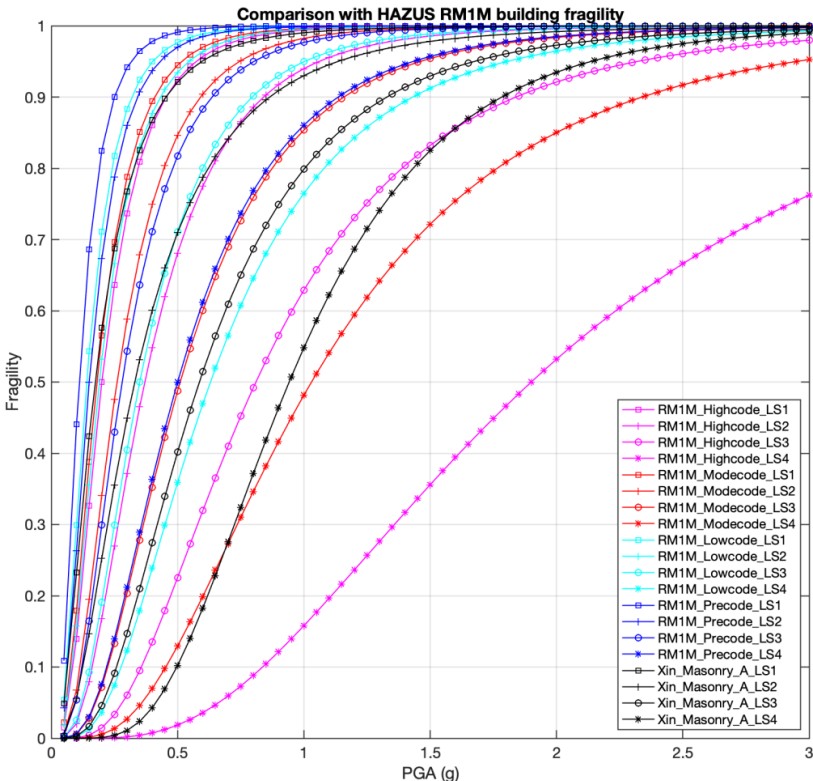

**Figure A6:** Median fragility curve comparison between HAZUS "RM1M" building type and our work for "Masonry_A". In HAZUS project, "RM1M" refers to "Mid-rise Reinforced Masonry Bearing Walls with Wood or Metal Deck Diaphragms" (see Sect. 4.2.4 for more details).


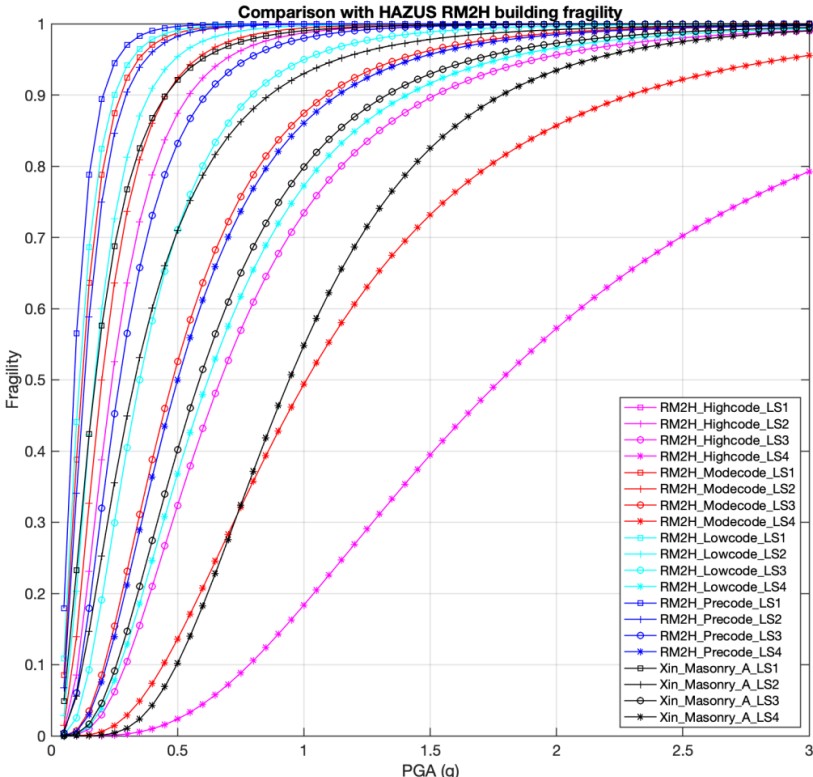

**Figure A7:** Median fragility curve comparison between HAZUS "RM2H" building and our work for "Masonry_A". In HAZUS project, "RM2H" refers to "High-rise Reinforced Masonry Bearing Walls with Precast Concrete Diaphragms" (see Sect. 4.2.4 for more details).

**Table 1:** Example of major damage states classification methods (modified after Rossetto and Elnashai, 2003).

| vulnerability | HAZUS1999 | EMS1998 | MSK1969 | AIJ1995 | China2008 |
|---|---|---|---|---|---|
| 0 | no damage | | | | |
| 10 | slight damage | Grade 1 | D1 | Light | D1 |
| 20 | | | | | D2 |
| 30 | | Grade 2 | D2 | Minor | D3 |
| 40 | | | | | |
| 50 | moderate damage | Grade 3 | D3 | Moderate | D4 |
| 60 | | | | | |
| 70 | | | | | |
| 80 | extensive damage | Grade 4 | D4 | Major | D5 |
| 90 | | | | | |
| 100 | complete damage | Grade 5 | | Partial collapse | |

**Table 2:** Detailed definition of building damage states in GB17742-2008, China.

| Damage state | Structural damage | Non-structural damage | Performance-based Description |
|---|---|---|---|
| D1 | Negligible | Cracks only in *very few* non-structural components | No need to repair, instant use |
| D2 | *Very few* components have visible cracks | Obvious cracks can be found | No need to repair or after slightly repairing, can be used directly |
| D3 | *A few* components have slight cracks, *very few* have obvious cracks | *Most* components have serious damage | Certain repair work should be done before continued use |
| D4 | *Most* components have serious damage, *a majority* have obvious cracks | *Most* components partially destroyed | The damage is difficult to repair |
| D5 | *The majority* components have serious damage, the building structure is close to collapse or already collapsed | Non-structural components are *commonly* destroyed | To repair the building back to normal is impossible |

5     Notes about qualifiers: "very few": <10%; "a few": 10%-50%; "most": 50%-70%; "majority": 70%-90%; "commonly": >90%.

**Table 3:** Divisions of seismic design level for Chinese buildings (modified after Lin et al., 2010).

| Seismic Resistance Design Level (PGA) | Construction Age | | | |
|---|---|---|---|---|
| | before 1978 | 1979-1989 | 1989-2001 | After 2001 |
| IX (0.4g) | pre-code | moderate-code | high-code | high-code |
| VIII (0.3g) | pre-code | moderate-code | moderate-code | high-code |
| VIII (0.2g) | pre-code | low-code | moderate-code | high-code |
| VII (0.15g) | pre-code | low-code | low-code | moderate-code |
| VII (0.10g) | pre-code | pre-code | low-code | low-code |
| VI (0.05g) | pre-code | pre-code | pre-code | low-code |





**Table 4:** The median fragility curve parameters regressed from empirical and analytical fragility data.

| data_source | build_type | fort _level | damage_state | $\mu_{LS}$ | $\sigma_{LS}$ | $R^2$ |
|---|---|---|---|---|---|---|
| Empirical | masonry | A | LS1 | 6.926 | 1.539 | 0.99 |
| | | | LS2 | 8.418 | 1.378 | 1 |
| | | | LS3 | 9.412 | 1.189 | 1 |
| | | | LS4 | 10.57 | 1.298 | 1 |
| | | B | LS1 | 7.658 | 1.393 | 0.98 |
| | | | LS2 | 9.283 | 1.298 | 0.99 |
| | | | LS3 | 10.43 | 1.505 | 0.99 |
| | | | LS4 | 11.59 | 1.553 | 1 |
| | RC | A | LS1 | 7.779 | 1.304 | 1 |
| | | | LS2 | 9.057 | 0.9367 | 1 |
| | | | LS3 | 9.893 | 0.9269 | 1 |
| | | | LS4 | 10.95 | 0.9626 | 1 |
| | | B | LS1 | 8.135 | 1.191 | 1 |
| | | | LS2 | 9.511 | 1.067 | 1 |
| | | | LS3 | 10.54 | 0.8831 | 1 |
| | | | LS4 | 11.77 | 1.075 | 1 |
| Analytical | masonry | A | LS1 | 0.1732 | 0.7512 | 1 |
| | | | LS2 | 0.33 | 0.7512 | 1 |
| | | | LS3 | 0.5862 | 0.6383 | 0.99 |
| | | | LS4 | 0.9416 | 0.4983 | 0.97 |
| | | B | LS1 | 0.3499 | 0.7573 | 1 |
| | | | LS2 | 0.6743 | 0.8101 | 1 |
| | | | LS3 | 1.281 | 0.8125 | 1 |
| | | | LS4 | 2.595 | 0.8581 | 0.99 |
| | RC | A | LS1 | 0.223 | 0.6615 | **0.80** |
| | | | LS2 | 0.353 | 0.7699 | **0.77** |
| | | | LS3 | 0.694 | 0.6111 | **0.73** |
| | | | LS4 | 1.404 | 0.4818 | 0.98 |
| | | B | LS1 | 0.315 | 0.539 | 0.99 |
| | | | LS2 | 0.46 | 0.5269 | 0.99 |
| | | | LS3 | 0.811 | 0.346 | 0.95 |
| | | | LS4 | 1.374 | 0.1763 | 0.91 |

*Note: "**fort_level**" **A** & **B** represent the pre/low/moderate-code and high-code seismic resistance level, respectively; "**damage_state**" LS1, LS2, LS3, LS4 represent the four damage limit states: slight, moderate, serious-, collapse, respectively; "$\mu_{LS}$" and "$\sigma_{LS}$" are the regression parameters between intensity/PGA and the corresponding fragilities of each damage limit state; $R^2$ indicates the fitness quality of the regressed median fragility curve, as plotted in Fig. 7 and Fig.8.





**Table 5:** The mean PGA values derived from intensity-PGA relations of "Masonry_A" based on the newly proposed approach (Sect. 5.4).

| intensity | VI | VII | VIII | IX | X |
|---|---|---|---|---|---|
| PGA(g) | 0.1 | 0.16 | 0.3 | 0.48 | 0.78 |

**Table 6:** The PGA ranges derived from more intensity-PGA relations (Sect. 5.5).

| intensity | VI | VII | VIII | IX | X |
|---|---|---|---|---|---|
| PGA(g) | 0.06-0.14 | 0.12-0.25 | 0.21-0.43 | 0.36-0.73 | 0.58-1.25 |

**Table 7:** The recommended intensity-PGA relations in China (GB17742-2008/1980).

| intensity | | VI | VII | VIII | IX | X |
|---|---|---|---|---|---|---|
| PGA | mean | 0.06 | 0.13 | 0.25 | 0.5 | 1.0 |
| (g) | range | 0.05-0.09 | 0.09-0.18 | 0.18-0.35 | 0.35-0.7 | 0.7-1.4 |

**Table 8:** The latest intensity-PGA relation derived by traditional practice for mainland China (Ding, 2017).

| intensity | | VI | VII | VIII | IX |
|---|---|---|---|---|---|
| PGA | mean | 0.09 | 0.16 | 0.3 | 0.55 |
| (g) | range | 0.06-0.12 | 0.09-0.22 | 0.22-0.41 | 0.41-0.75 |




**Table B1:** Statistics of fragility database for each damage limit state and each building type.

| data source | build_type | intensity/ PGA(g) | original fragility number | fragility number after removing outliers | | | | median value of each fragility dataset with truncated exceed. prob. ≥ 1% | | | | standard deviation of each fragility dataset with truncated median exceed. prob. ≥ 1% | | | |
|---|---|---|---|---|---|---|---|---|---|---|---|---|---|---|---|
| | | | | LS1 | LS2 | LS3 | LS4 | LS1 | LS2 | LS3 | LS4 | LS1 | LS2 | LS3 | LS4 |
| empirical | Masonry_A | 6 | 29 | 28 | 28 | 28 | 28 | 0.30 | 0.06 | 0.01 | | 0.13 | 0.04 | 0.01 | |
| | | 7 | 29 | 29 | 26 | 26 | 27 | 0.47 | 0.14 | 0.04 | | 0.21 | 0.08 | 0.04 | |
| | | 8 | 29 | 29 | 29 | 25 | 26 | 0.78 | 0.40 | 0.11 | 0.03 | 0.20 | 0.27 | 0.07 | 0.02 |
| | | 9 | 28 | 28 | 28 | 28 | 25 | 0.91 | 0.64 | 0.36 | 0.11 | 0.14 | 0.27 | 0.27 | 0.06 |
| | | 10 | 28 | 27 | 26 | 28 | 28 | 0.99 | 0.90 | 0.69 | 0.33 | 0.05 | 0.15 | 0.26 | 0.22 |
| | Masonry_B | 6 | 21 | 21 | 21 | 21 | 21 | 0.15 | 0.02 | | | 0.09 | 0.02 | | |
| | | 7 | 21 | 21 | 20 | 18 | 18 | 0.26 | 0.08 | 0.02 | | 0.21 | 0.10 | 0.03 | |
| | | 8 | 21 | 21 | 21 | 21 | 18 | 0.66 | 0.17 | 0.07 | 0.01 | 0.30 | 0.24 | 0.16 | 0.01 |
| | | 9 | 20 | 20 | 20 | 20 | 17 | 0.79 | 0.37 | 0.15 | 0.05 | 0.29 | 0.30 | 0.25 | 0.03 |
| | | 10 | 20 | 20 | 20 | 20 | 20 | 0.96 | 0.74 | 0.39 | 0.15 | 0.24 | 0.31 | 0.30 | 0.22 |
| | RC_A | 6 | 24 | 23 | 22 | 19 | 24 | 0.12 | | | | 0.07 | | | |
| | | 7 | 24 | 23 | 23 | 22 | 24 | 0.25 | 0.02 | | | 0.14 | 0.05 | | |
| | | 8 | 26 | 26 | 24 | 24 | 23 | 0.57 | 0.12 | 0.02 | | 0.19 | 0.12 | 0.06 | |
| | | 9 | 20 | 20 | 20 | 19 | 18 | 0.82 | 0.48 | 0.17 | 0.02 | 0.14 | 0.17 | 0.08 | 0.02 |
| | | 10 | 16 | 16 | 16 | 16 | 14 | 0.98 | 0.84 | 0.55 | 0.16 | 0.10 | 0.16 | 0.26 | 0.10 |
| | RC_B | 6 | 6 | 6 | 5 | 6 | 6 | 0.05 | | | | 0.05 | | | |
| | | 7 | 6 | 5 | 5 | 6 | 6 | 0.15 | 0.02 | | | 0.06 | 0.01 | | |
| | | 8 | 6 | 6 | 5 | 5 | 6 | 0.48 | 0.06 | | | 0.19 | 0.02 | | |
| | | 9 | 5 | 5 | 5 | 5 | 5 | 0.75 | 0.33 | 0.04 | | 0.20 | 0.18 | 0.11 | |
| | | 10 | 5 | 5 | 5 | 5 | 5 | 0.95 | 0.67 | 0.27 | 0.05 | 0.15 | 0.25 | 0.29 | 0.11 |
| analytical | Masonry_A | 0.1 | 6 | 6 | 6 | 5 | 6 | 0.22 | 0.06 | 0.02 | | 0.14 | 0.06 | 0.01 | |
| | | 0.2 | 6 | 6 | 6 | 6 | 6 | 0.60 | 0.25 | 0.08 | 0.02 | 0.15 | 0.13 | 0.06 | 0.01 |
| | | 0.3 | 6 | 6 | 6 | 6 | 6 | 0.77 | 0.47 | 0.18 | 0.05 | 0.10 | 0.14 | 0.10 | 0.03 |
| | | 0.4 | 6 | 6 | 6 | 6 | 6 | 0.86 | 0.60 | 0.29 | 0.09 | 0.07 | 0.14 | 0.14 | 0.07 |
| | | 0.5 | 6 | 6 | 6 | 6 | 6 | 0.92 | 0.70 | 0.39 | 0.14 | 0.05 | 0.12 | 0.16 | 0.10 |
| | | 0.6 | 6 | 6 | 6 | 6 | 6 | 0.95 | 0.77 | 0.50 | 0.20 | 0.04 | 0.10 | 0.16 | 0.13 |
| | | 0.7 | 6 | 6 | 6 | 6 | 6 | 0.97 | 0.84 | 0.59 | 0.27 | 0.03 | 0.08 | 0.15 | 0.15 |
| | | 0.8 | 6 | 6 | 6 | 6 | 6 | 0.98 | 0.88 | 0.66 | 0.34 | 0.02 | 0.07 | 0.15 | 0.17 |
| | | 0.9 | 6 | 6 | 6 | 6 | 6 | 0.99 | 0.91 | 0.73 | 0.41 | 0.02 | 0.06 | 0.15 | 0.18 |
| | | 1 | 6 | 6 | 6 | 6 | 6 | 0.99 | 0.94 | 0.78 | 0.47 | 0.02 | 0.05 | 0.14 | 0.19 |
| | | 1.1 | 2 | 2 | 2 | 2 | 2 | 1.00 | 0.97 | 0.91 | 0.70 | 0.00 | 0.03 | 0.08 | 0.18 |
| | | 1.2 | 2 | 2 | 2 | 2 | 2 | 1.00 | 0.98 | 0.93 | 0.74 | 0.00 | 0.03 | 0.09 | 0.16 |
| | Masonry_B | 0.1 | 6 | 6 | 6 | 6 | 6 | 0.04 | 0.02 | | | 0.05 | 0.01 | | |
| | | 0.2 | 6 | 6 | 6 | 6 | 5 | 0.21 | 0.05 | 0.01 | | 0.14 | 0.05 | 0.01 | |
| | | 0.3 | 6 | 6 | 6 | 6 | 5 | 0.43 | 0.14 | 0.04 | | 0.19 | 0.09 | 0.02 | |
| | | 0.4 | 6 | 6 | 6 | 6 | 6 | 0.59 | 0.25 | 0.08 | 0.01 | 0.21 | 0.14 | 0.05 | 0.01 |
| | | 0.5 | 6 | 6 | 6 | 6 | 6 | 0.69 | 0.37 | 0.13 | 0.03 | 0.20 | 0.17 | 0.07 | 0.02 |
| | | 0.6 | 6 | 6 | 6 | 6 | 6 | 0.76 | 0.45 | 0.18 | 0.05 | 0.17 | 0.18 | 0.09 | 0.03 |
| | | 0.7 | 6 | 6 | 6 | 6 | 6 | 0.81 | 0.53 | 0.22 | 0.07 | 0.14 | 0.18 | 0.11 | 0.04 |
| | | 0.8 | 6 | 5 | 6 | 6 | 6 | 0.86 | 0.59 | 0.28 | 0.09 | 0.06 | 0.17 | 0.13 | 0.06 |
| | | 0.9 | 6 | 5 | 6 | 6 | 6 | 0.89 | 0.65 | 0.33 | 0.11 | 0.05 | 0.16 | 0.14 | 0.08 |
| | | 1 | 6 | 5 | 6 | 6 | 6 | 0.91 | 0.70 | 0.39 | 0.13 | 0.04 | 0.15 | 0.15 | 0.10 |
| | | 1.1 | 3 | 3 | 3 | 3 | 3 | 0.93 | 0.70 | 0.42 | 0.15 | 0.09 | 0.20 | 0.23 | 0.17 |
| | | 1.2 | 3 | 3 | 3 | 3 | 3 | 0.95 | 0.75 | 0.48 | 0.19 | 0.08 | 0.19 | 0.24 | 0.19 |
| | RC_A | 0.1 | 20 | 18 | 18 | 20 | 17 | 0.07 | | | | 0.07 | | | |
| | | 0.2 | 20 | 20 | 18 | 19 | 20 | 0.42 | 0.13 | 0.01 | | 0.32 | 0.12 | 0.03 | |
| | | 0.3 | 22 | 22 | 22 | 21 | 20 | 0.72 | 0.45 | 0.05 | | 0.29 | 0.35 | 0.09 | |
| | | 0.4 | 20 | 20 | 20 | 20 | 18 | 0.78 | 0.48 | 0.10 | 0.02 | 0.26 | 0.36 | 0.21 | 0.02 |
| | | 0.5 | 13 | 12 | 13 | 13 | 11 | 0.96 | 0.89 | 0.34 | 0.03 | 0.09 | 0.25 | 0.26 | 0.04 |
| | | 0.6 | 22 | 22 | 22 | 22 | 19 | 0.93 | 0.82 | 0.33 | 0.05 | 0.30 | 0.41 | 0.32 | 0.05 |
| | | 0.7 | 11 | 11 | 11 | 11 | 10 | 0.99 | 0.96 | 0.77 | 0.06 | 0.08 | 0.22 | 0.33 | 0.06 |
| | | 0.8 | 17 | 17 | 17 | 17 | 15 | 0.88 | 0.64 | 0.37 | 0.15 | 0.38 | 0.36 | 0.37 | 0.08 |
| | | 0.9 | 12 | 11 | 12 | 12 | 11 | 1.00 | 0.99 | 0.92 | 0.14 | 0.03 | 0.16 | 0.30 | 0.11 |
| | | 1 | 16 | 16 | 16 | 16 | 15 | 0.91 | 0.70 | 0.41 | 0.25 | 0.49 | 0.38 | 0.34 | 0.13 |
| | | 1.1 | 5 | 5 | 5 | 5 | 5 | 1.00 | 0.99 | 0.99 | 0.29 | 0.00 | 0.01 | 0.03 | 0.31 |
| | | 1.2 | 14 | 14 | 14 | 14 | 14 | 0.61 | 0.68 | 0.67 | 0.39 | 0.47 | 0.43 | 0.34 | 0.27 |
| | RC_B | 0.1 | 9 | 8 | 9 | 9 | 9 | 0.02 | | | | 0.02 | | | |
| | | 0.2 | 9 | 8 | 7 | 9 | 9 | 0.18 | 0.04 | | | 0.28 | 0.02 | | |
| | | 0.3 | 11 | 11 | 11 | 10 | 11 | 0.50 | 0.22 | | | 0.35 | 0.32 | | |
| | | 0.4 | 9 | 9 | 9 | 8 | 9 | 0.65 | 0.37 | 0.04 | | 0.25 | 0.33 | 0.04 | |
| | | 0.5 | 9 | 9 | 9 | 8 | 8 | 0.79 | 0.57 | 0.08 | | 0.21 | 0.31 | 0.08 | |
| | | 0.6 | 11 | 11 | 11 | 10 | 10 | 0.93 | 0.75 | 0.20 | 0.02 | 0.18 | 0.30 | 0.10 | 0.01 |
| | | 0.7 | 9 | 9 | 9 | 9 | 8 | 0.93 | 0.81 | 0.37 | 0.03 | 0.15 | 0.29 | 0.22 | 0.02 |
| | | 0.8 | 8 | 8 | 8 | 8 | 7 | 0.91 | 0.79 | 0.45 | 0.03 | 0.12 | 0.26 | 0.29 | 0.02 |
| | | 0.9 | 10 | 10 | 10 | 10 | 9 | 0.99 | 0.93 | 0.68 | 0.03 | 0.12 | 0.25 | 0.33 | 0.03 |
| | | 1 | 7 | 7 | 7 | 7 | 7 | 0.94 | 0.83 | 0.52 | 0.05 | 0.09 | 0.22 | 0.33 | 0.04 |
| | | 1.1 | 4 | 4 | 4 | 4 | 4 | 1.00 | 0.97 | 0.89 | 0.08 | 0.02 | 0.05 | 0.19 | 0.07 |
| | | 1.2 | 6 | 5 | 5 | 5 | 6 | 1.00 | 0.99 | 0.98 | 0.23 | 0.00 | 0.01 | 0.06 | 0.12 |



Note: "origin fragility number" refers to the number of original fragilities collected for each damage limit state of each building type from previous studies; "fragility number after removing outliers" refers to the remaining fragilities after removing outliers using box-plot check method. Only intensity and PGA values with truncated exceedance probability ≥1% for each damage limit state of each building type are given, since higher damage states can appear only for higher intensities 5 or PGA values (see Sect. 5.2 for more details).



**Table B2:** Chinese Official Seismic Intensity Scale: GB17742-2008 (modified after CSIS, 2019).

| Macro Intensity | Senses by people on the ground | Degree of building damage | | | Other damages | Horizontal motion on the ground | |
|---|---|---|---|---|---|---|---|
| | | Building type | Damages | Average damage index | | Peak acceleration(m/s²) | Peak speed (m/s) |
| I | Insensible | | | | | | |
| II | Sensible by very few still indoor people | | | | | | |
| III | Sensible by a few still indoor people | | Slight rattle of doors and windows | | Slight swing of suspended objects | | |
| IV | Sensible by most people indoors, a few people outdoors; a few wake up from sleep | | Rattle of doors and windows | | Obvious swing of suspended objects; vessels rattle | | |
| V | Commonly sensible by people indoors, sensible by most people outdoors; most wake up from sleep | | Noise from vibration of doors, windows, and building frames; falling of dusts, small cracks in plasters, falling of some roof tiles, bricks falling from a few roof-top chimneys | | Rocking or flipping of unstable objects | 0.31 (0.22-0.44) | 0.03 (0.02-0.04) |
| VI | Most unable to stand stably, a few scared to running outdoors | A | A few have D3 damage | 0-0.11 | Cracks in river banks and soft soil; occasional burst of sand and water from saturated sand layers; cracks on some standalone chimneys | 0.63 (0.45-0.89) | 0.06 (0.05-0.09) |
| | | B | Very few have D3 damage, a few have D2 damage, most are intact | | | | |
| | | C | Very few have D2 damage, the majority are intact | 0-0.08 | | | |
| VII | Majority scared to running outdoors, sensible by bicycle riders and people in moving motor vehicles | A | A few have D4 and/or D5 damage, most have D3 and/or D2 damage | 0.09-0.31 | Collapse of river banks; frequent burst of sand and water from saturated sand layers; many cracks in soft soils; moderate destruction of most standalone chimneys | 1.25 (0.90-1.77) | 0.13 (0.10-0.18) |
| | | B | A few have D3 damage, most have D2 and/or D1 damage | | | | |
| | | C | A few have D3 and/or D2, most are intact | 0.07-0.22 | | | |
| VIII | Most swing about, difficult to walk | A | A few have D5 damage, most have D4 and/or D3 damage | 0.29-0.51 | Cracks appear in hard dry soils; severe destruction of most standalone chimneys; tree tops break; death of people and cattle caused by building destruction | 2.50 (1.78-3.53) | 0.25 (0.19-0.35) |
| | | B | Very few have D5 damage, most have D3 and/or D2 damage | | | | |
| | | C | A few have D4 and/or D3 damage, most have D2 damage | 0.2-0.4 | | | |
| IX | Moving people fall | A | Most have D4 and/or D5 damage | 0.49-0.71 | Many cracks in hard dry soils; possible cracks and dislocations in bedrock; frequent landslides and collapses; collapse of many standalone chimneys | 5.00 (3.54-7.07) | 0.50 (0.36-0.71) |
| | | B | A few have D5 damage, most have D4 and/or D3 damage | | | | |
| | | C | A few have D5 and/or D4 damage, most have D3 and/or D2 damage | 0.38-0.6 | | | |
| X | Bicycle riders may fall; people in unstable state may fall away; | A | Commonly have D5 damage | 0.69-0.91 | Cracks in bedrock and earthquake fractures; destruction of bridge arches | 10.00 (7.08-14.14) | 1.00 (0.72-1.41) |
| | | B | The majority have D5 damage | | | | |





| | sense of being thrown up | C | Most have D5 and/or D4 damage | 0.58-0.8 | founded in bedrock; foundation damage or collapse of most standalone chimneys | | |
|---|---|---|---|---|---|---|---|
| **XI** | | A | Commonly have D5 damage | 0.89–1.0 | Earthquake fractures extend a long way; many bedrock cracks and landslides | | |
| | | B | | | | | |
| | | C | | 0.78-1.0 | | | |
| **XII** | | A | Almost all have D5 damage | 1.0 | Drastic change in landscape, mountains, and rivers | | |
| | | B | | | | | |
| | | C | | | | | |

Notes about Qualifiers: "very few": <10%; "few": 10% - 50%; "most": 50% - 70%; "majority": 70% - 90%; "commonly": >90%.





**Appendix C: Methodology in characterization of uncertainty transmission from empirical/analytical fragility database to intensity-PGA relation**

The estimation of the uncertainty of the intensity-PGA relation (Eq. (5)) is not a standard procedure like regression analysis. We have fragility as function of intensity with an error on the fragility so that fragility is a random variable. It is also a random variable when derived as function of $y = \ln(PGA)$. We express this as

$$f(y) = g(y) + \varepsilon_g \tag{C1}$$

$$f(i) = h(i) + \varepsilon_h \tag{C2}$$

With $i$: intensity, $y$: ln(PGA), $f$: fragility.

$\varepsilon_g$ is a normally distributed random variable with zero mean, standard deviation $\sigma_g$.

$\varepsilon_h$ is a normally distributed random variable with zero mean, standard deviation $\sigma_h$.

$g(y)$ and $h(i)$ are non-linear functions that can be modelled as cumulative normal distributions in intensity and ln(PGA) as fragility ranges between 0 and 1. Under this condition equating the expectation values of the fragilities

$$E[f(y)] = E[f(i)], g(y) = h(i) \tag{C3}$$

Leads to a linear relation between ln(PGA) and intensity. Including uncertainties in this relation leads to the hypothesis

$$\ln(PGA) = y = \alpha + \beta \cdot i + \varepsilon_y \tag{C4}$$

$\varepsilon_y$ is a normally distributed random variable with zero mean, standard deviation $\sigma_y$ and this is the quantity we want to determine. Note that with this relation y became a random variable. Its expectation value is related to intensity via

$$E[y] = \bar{y} = \alpha + \beta \cdot i \tag{C5}$$

We ask the question: If the above relation holds and intensity is fixed what range of values for y is possible so that

$$f(y(i)) = f(i) \tag{C6}$$

holds. Inserting above expressions provides

$$g(\alpha + \beta \cdot i + \varepsilon_y) + \varepsilon_g = h(i) + \varepsilon_h \tag{C7}$$

If we assume that the error term is small, we can write:

$$g(\alpha + \beta \cdot i + \varepsilon_y) \approx g(\alpha + \beta \cdot i) + g'(\alpha + \beta \cdot i) \cdot \varepsilon_y \tag{C8}$$




$g'(\alpha + \beta \cdot i)$ is the slope of the g(y) curve and has the unit 1/ln(PGA). The value changes along the curve so that we replace it by an average value $\bar{g}'$. Then,

$$\varepsilon_y = \frac{1}{\bar{g}'}(\varepsilon_h - \varepsilon_g) \tag{C9}$$

and under the assumption of independence of the two random terms we get

$$\sigma_y = \frac{1}{\bar{g}'}\sqrt{\sigma_h^2 + \sigma_g^2} \tag{C10}$$

In order to utilize this estimation scheme for our data we approximate $\bar{g}'$ by its value at the 0.5 value of the fragility function: $g(y_m) = 0.5$, so that $\bar{g}' = g'(y_m)$. When we do the estimates for each damage class and each building type we find the standard deviations for ln(PGA) according to the following table. The values do vary. A representative/average value appears to be 0.3.

**Table B3**: The standard deviation in intensity-PGA relation for each damage limit state of each building type.

| Build_Type | LS1 | LS2 | LS3 | LS4 |
|---|---|---|---|---|
| Masonry_A | 0.29 | 0.34 | 0.27 | 0.20 |
| Masonry_B | 0.48 | 0.49 | 0.44 | 0.25 |
| RC_A | 0.44 | 0.59 | 0.42 | 0.16 |
| RC_B | 0.27 | 0.32 | 0.24 | 0.05 |