# Peer review of "Review of fragility analyses for major building types in China with new implications for intensity-PGA relation development"

_Natural Hazards and Earth System Sciences, 2019_

## Referee Comment (RC1) · Mustafa Erdik (Referee) · 7 Sep 2019

The paper essentially aims the development of intensity-PGA relationships using a novel method that relies on comparison of intensity-based empirical and PGA-based analytical fragility relationships for the same building types from China. To fulfill this object, the authors first review the empirical building fragility database, mostly for China, scrutinize the data and derive the median Chinese intensity-based fragility relationships with basic treatment of uncertainties. For this empirical fragility study, three types of masonry buildings with different construction practices are considered. Secondly, the authors inspected publications that provide PGA-based analytical fragility functions

(dependent on PGA) for the same damage classes and building categories. Thus, a solid fragility database and median fragility relationships, based on both intensity and PGA, areas established for mainland China. For the derivation of median fragility relationships, the lognormal distribution is used with excellent goodness of fit. The paper culminates with the description and application of the novel approach for the development of the intensity-PGA relation by using fragility as the transfer medium. The results obtained are very valuable and compare well with limited relationships based on direct regression of measured PGA with the assessed intensity values. The specific issues with the paper can be listed as follows: • Very comprehensive literature review and description of ingredients and methodology on the assessment of fragility relationships from empirical data. • Text can be shortened since involves several repetitions of the objectives and methodologies. • With the exception of the information provided on general approaches on the derivation of analytical vulnerabilities, not much detail is provided on the papers that the PGA-based analytical fragilities for the Chinese building stock. It appears that, with the exception of outlier removal, results on all these papers are given the same weight for the median fragility assessment. • The uncertainties in the fragility assessments are not adequately covered in the paper with the exception of uncertainties illustrated in Appendix Fig. A1-A4 and Table B1). • Direct comparison of different fragility relationships is a difficult issue due to different building, damage state and ground motion intensity definitions and attributes considered in these relationships. This fact also manifests itself in this paper. Upon comparison of fragility relationships obtaÄśned in this paper with the results of several relevant international projects, only one (HAZUS Project) similarity for "Masonry – A" building type was found. • The intensity-PGA relationships developed by using the correspondence between the empirical and PGA-based analytical fragility relationships is based on a novel approach and would be very valuable for use in international projects. However, a description on the relationship between the Chinese Official Seismic Intensity Scale (GB17742) and the other internationally adopted scales (e.g. MMI, MSK, EMS) nay need to be included (or referenced) in the paper. • The methodology in the

transmission of uncertainty from empirical/analytical fragility database to the intensity-PGA relation is provided in Appendix C. This transmission of uncertainty is important and should preferably be integrated into the main text of the paper.

---

## Referee Comment (RC2) · Anonymous Referee #2 · 8 Nov 2019

This is a very interesting paper tying the fragility curves to vulnerability and through this get a better correlation into conversion of Intensities to PGA. The authors reviewed a large number of data published on Chinese earthquakes since 1975 for two main typologies: RC and Masonry: 69 in terms of Chinese Intensity Scale and 18 on PGA. They used fragility curves obtained from experimental (in function of INT) and from numerical developments (in function of PGA) to obtain a conversion between INT vs PGA. To expand the results obtained to other places around the World, and not only to the Chinese construction, in a tentative to propose a world-wide conversion, they compare the results with other proposals in Europe and US (PERPETUATE; SHARE; GEM; etc.) But before I get convinced with the results presented and consequently to accept

the paper I need a response to a few important issues. 1- Construction in China is very different from other regions. So the comparison with other regions is very difficult to accept. 2- Only two Types RC and Masonry (sub-divided into two sub classes) are considered. EMS-98 has many more! 3- The final proposal (TableB2) for each Intensity gets an interval 1:2 and no results for I>X. Other studies show similar intervals but with centre value slightly deviated. 4- Errors on establishing the D1 to D4 or D5 are full of uncertainties and so I do not know if bringing the fragilities into place is reducing the error. 5- The Indicator with people and bicycles is very interesting especially for China as there is much traffic with bikes. For other countries unfortunately, the situation is still too far! Few Problems to correct: 1- There should be a Figure representing graphically the results in Table B2 2- Formula (5) is very old in California 3- Fig 3 and 4 should have line to be more understandable. 4- Fig 5 and 5 should have line and same colour. 5- Fig A1 could be merged with Fig 7 and Fig A2 with Fig 8. 6- Figs A6 and A7 are impossible to read! Title could be slighted changed, inverting the order of ideas.

I found several inconsistencies in the Reference List Attached Doc (References Corrections)

Please also note the supplement to this comment:
https://www.nat-hazards-earth-syst-sci-discuss.net/nhess-2019-195/nhess-2019-195-RC2-supplement.pdf

**Supplement:**

[Figure]

[Figure]

**References**

**In English:**

Antoniou, S. and Pinho, R.: Development and verification of a displacement-based adaptive pushover procedure, Journal of Earthquake Engineering, 8, 643-661, https://doi.org/10.1080/13632460409350504, 2004.

5   Bilal, M. and Askan, A.: Relationships between Felt Intensity and Recorded Ground-Motion Parameters for Turkey, B SEISMOL SOC AM, 104, 484-496, https://doi.org/10.1785/0120130093, 2014.

Billah, A. H. M. M. and Alam, M. S.: Seismic fragility assessment of highway bridges: A state-of-the-art review, Structure and Infrastructure Engineering, 11, 804-832, https://doi.org/10.1080/15732479.2014.912243, 2015.

Calvi, G. M., Pinho, R. and Magenes, G.: Development of seismic vulnerability assessment methodologies over t
10  he past 30 years, ISET journal of Earthquake Technology, 43, 75-104, https://www.researchgate.net/publication/241826044, 2006.

Caprio, M., Tarigan, B. and Worden, C. B.: Ground motion to intensity conversion equations (GMICEs): A global relationship and evaluation of regional dependency, Bulletin of the Seismological Society of America, 105, 1476-1490, https://doi.org/10.1785/0120140286, 2015.

15  CSIS: China seismic intensity scale, a non-official English translation based on contents in Wikipedia, https://en.wikipedia.org/w/index.php?title=China_seismic_intensity_scale&oldid=812457225, last access: 21 May 2019.

Crowley, H., Colombi, M., Silva, V., Ahmad N., Fardis M., Tsionis G., Papailia A., Taucer F., Hancilar U., Yakut A., Erberik M.A.: D3.1. Fragility functions for common RC building types in Europe, Pavia, Italy, 223
20  pp., http://www.vce.at/SYNER-G/files/dissemination/deliverables.html, 2011a.

Crowley, H., Colombi M., Silva, V., Ahmad N., Fardis M., Tsionis G., Karatoni T., Lyrantazaki F., Taucer F., Hancilar U., Yakut A., Erberik M.A.: D3.2. Fragility functions for common masonry building types in Europe, Pavia, Italy, 177 pp., http://www.vce.at/SYNER-G/files/dissemination/deliverables.html, 2011b.

Draper, N. R., Smith, H: Applied Regression Analysis, 3rd ed., John Wiley & Sons, New York, United States,
25  2014.

Del Gaudio, C., Ricci, P. and Verderame, G. M.: Development and urban-scale application of a simplified method for seismic fragility assessment of RC buildings, Engineering Structures, 91, 40-57, https://doi.org/10.1016/j.engstruct.2015.01.031, 2015.

Dumova-Jovanoska, E.: Fragility curves for reinforced concrete structures in Skopje (Macedonia) region, Soil
30  Dynamics and Earthquake Engineering, 19, 455-466, https://doi.org/10.1016/S0267-7261(00)00017-8, 2000.

EMS1998: European Macro-seismic Scale 1998, European Seismological Commission, sub commission on Engineering Seismology, Working Group, Macro-seismic Scales, Conseil de l'Europe, Cahiers du Centre Européen de Géodynamique et de Séismologie, Vol. 15, Luxembourg, 1998.

Fajfar, P.: A nonlinear analysis method for performance-based seismic design, Earthquake Spectra, 16, 573-592,
35  https://doi.org/10.1193/1.1586128, 2000.

FEMA: HAZUS®MH estimated annualized earthquake losses for the United States, FEMA 366, Washington DC, United States, https://secure.madcad.com/media/fema/FEMA-366-2008.pdf, 2008.

FEMA: Multi-hazard loss estimation methodology: earthquake model (HAZUS-MH-MR3), Technical Report, Washington DC, USA, https://www.fema.gov/media-library/assets/documents/24609, 2003.

40  Freeman, S. A.: The capacity spectrum method, in Proceedings of the 11th European Conference on Earthquake Engineering, Paris., http://citeseerx.ist.psu.edu/viewdoc/summary?doi=10.1.1.460.2405, 1998.

Freeman, S. A.: Review of the development of the capacity spectrum method, ISET Journal of Earthquake Technology, 41(1), 1–13, http://citeseerx.ist.psu.edu/viewdoc/download?doi=10.1.1.451.9286, 2004.

---

## Author Comment (AC1) · 17 Nov 2019

The authors are grateful to the time spent by Prof. Mustafa Erdik (hereafter referred to as "RC1") on carefully reading our work and giving a series of constructive comments. Detailed responses to these comments are given as follows. *The comments are in purple with italic font* and our responses are in blue.

(i) General comments from RC1:

*"The paper essentially aims the development of intensity-PGA relationships using a novel method that relies on comparison of intensity-based empirical and PGA-based analytical fragility relationships for the same building types from China. To fulfill this object, the authors first review the empirical building fragility database, mostly for China, scrutinize the data and derive the median Chinese intensity-based fragility relationships with basic treatment of uncertainties. For this empirical fragility study, three types of masonry buildings with different construction practices are considered. Secondly, the authors inspected publications that provide PGA-based analytical fragility functions (dependent on PGA) for the same damage classes and building categories. Thus, a solid fragility database and median fragility relationships, based on both intensity and PGA, are established for mainland China. For the derivation of median fragility relationships, the lognormal distribution is used with excellent goodness of fit. The paper culminates with the description and application of the novel approach for the development of the intensity-PGA relation by using fragility as the transfer medium. The results obtained are very valuable and compare well with limited relationships based on direct regression of measured PGA with the assessed intensity values. "*

(ii) Response:

Thank you for this summarization and such positive judgement on our work.

(i) RC1 comment 1:

*"Very comprehensive literature review and description of ingredients and methodology on the assessment of fragility relationships from empirical data."*

(ii) Response: Thank you for saying so.

(i) RC1 comment 2:

*" Text can be shortened since involves several repetitions of the objectives and methodologies."*

(ii) Response: From the feedbacks of previous reviewing process, some reviewers misunderstood or partially neglected the focuses of this work. Therefore, although the objectives and methodologies are firstly mentioned in the Introduction section, to emphasize and to avoid possible misunderstanding from future readers, the focuses are thus reiterated in Section 4 and Section 5.2.

(i) RC1 comment 3:

*"With the exception of the information provided on general approaches on the derivation of analytical vulnerabilities, not much detail is provided on the papers that the PGA-based analytical fragilities for the Chinese building stock. It appears that, with the exception of outlier removal, results on all these papers are given the same weight for the median fragility assessment. "*

(ii) Response: The analytical fragility related studies for the Chinese building stock generally follow the classical methods and procedures as summarized in Page 4 Line27-36. Therefore, the details of these procedures are not presented in detail, since there is marginal exception.

You are right in that we do give the same weight for different intensity/PGA levels in regressing the median fragility curves, to avoid the introducing of extra subjective uncertainty. Since we noticed in previous literature, some researchers gave higher weight to lower intensity/PGA levels (e.g. Ding et al., 2017) in their regression, while others gave higher weight to higher intensity/PGA levels (e.g. Ma et al., 2014) with different focuses.

(i) RC1 comment 4:

*"The uncertainties in the fragility assessments are not adequately covered in the paper with the exception of uncertainties illustrated in Appendix Fig. A1-A4 and Table B1)."*

(ii) Response: Thank you for pointing this out. RC2 also gives a detailed suggestion in regard of this. We'll accept the suggestions of you two and combine the error-bar analysis in the Appendix Fig. A1 and Fig. A2 with the median fragility curve in Fig. 7 and Fig. 8 in the main context, respectively.

(i) RC1 comment 5:

*"Direct comparison of different fragility relationships is a difficult issue due to different building, damage state and ground motion intensity definitions and attributes considered in these relationships. This fact also manifests itself in this paper. Upon comparison of fragility relationships obtained in this paper with the results of several relevant international projects, only one (HAZUS Project) similarity for "Masonry – A" building type was found."*

(ii) Response: Yes, this part was added to respond the comments of one previous reviewer. And we do notice the difficulty to conduct such comparisons, given the differences you summarized in the comment among different international projects. Such a grand topic deserves individual deep-going study in the future. Therefore, to keep the integrity and narrow down the focus of our current work, we'll remove the comparisons in Section 4.2 and delete the related descriptions in and main context as well as figures in the Appendix part.

(i) RC1 comment 6:

*"The intensity-PGA relationships developed by using the correspondence between the empirical and PGA-based analytical fragility relationships is based on a novel approach and would be very valuable for use in international projects. However, a description on the relationship between the Chinese Official Seismic Intensity Scale (GB17742) and the other internationally adopted scales (e.g. MMI, MSK, EMS) may need to be included (or referenced) in the paper."*

(ii) Response: Thank you for this suggestion. We found in previous studies (e.g. in Daniell, 2014) such work has been conducted. We'll add these references in the main context of the revised version.

[Figure]

Figure R1: Comparison of Intensity Scales in Daniell (2014, in his Figure 9), after the work of Gorshkov and Shenkareva (1960), Barosh (1969), Musson et al. (2010) (Note: in this figure, "Liedu-1980/1999" represents the Chinese Seismic Intensity Scale).

(i) RC1 comment 7:

*"The methodology in the transmission of uncertainty from empirical/analytical fragility database to the intensity-PGA relation is provided in Appendix C. This transmission of uncertainty is important and should preferably be integrated into the main text of the paper."*

(ii) Response: Thank you for this suggestion. But since the uncertainty of this newly derived intensity-PGA relationship is mentioned only as a number in Section 5.4, to keep the structure of this work clear and also to narrow down the focus, we still consider that it is better to put the uncertainty transmission methodology in the Appendix part for interested readers to have a further check.

**References:**

Barosh, P. J.: Use of seismic intensity data to predict the effects of earthquakes and underground nuclear explosions in various geologic settings, US Government Printing Office., 1969.

Daniell, J.: Development of socio-economic fragility functions for use in worldwide rapid earthquake loss estimation procedures, Ph.D. Thesis, Karlsruhe Institute of Technology, Karlsruhe, Germany., 2014.

Ding, B., Sun, J., Du, K. and Luo, huan: Study on relationships between seismic intensity and peak ground acceleration, peak ground velocit, Earthquake Engineering and Engineering Dynamics, 37(2), 26–36, http://dx.doi.org/%2010.13197/j.eeev.2017.02.26.dingbr.004, 2017.

Gorshkov, G. P. and Shenkareva, G. A.: On the Correlation of Seismic Scales -- USSR --, JOINT PUBLICATIONS RESEARCH SERVICE ARLINGTON VA. [online] Available from: https://apps.dtic.mil/docs/citations/ADA362451 (Accessed 14 November 2019), 1960.

Ma, Q., Li, S., Li, S. and Tao, D.: On the correlation of ground motion patterns with seismic intensity, Earthquake Engineering and Engineering Dynamics, 34(4), 83–92, doi:10.13197/j.eeev.2014.04.83.maq.011, 2014.

Musson, R. M., Grünthal, G. and Stucchi, M.: The comparison of macroseismic intensity scales, Journal of Seismology, 14(2), 413–428, 2010.

---

## Author Comment (AC2) · 17 Nov 2019

We deeply appreciate the detailed check of our work by the anonymous reviewer (hereafter referred as "RC2") and the helpful advices proposed. Detailed responses to RC2 comments are elaborated as follows. *The comments are in purple with italic font* and our responses are in blue.

(i) General comments from RC2:
*"This is a very interesting paper tying the fragility curves to vulnerability and through this get a better correlation into conversion of Intensities to PGA. The authors reviewed a large number of data published on Chinese earthquakes since 1975 for two main typologies: RC and Masonry: 69 in terms of Chinese Intensity Scale and 18 on PGA. They used fragility curves obtained from experimental (in function of INT) and from numerical developments (in function of PGA) to obtain a conversion between INT vs PGA. To expand the results obtained to other places around the World, and not only to the Chinese construction, in a tentative to propose a world-wide conversion, they compare the results with other proposals in Europe and US (PERPETUATE; SHARE; GEM; etc.). But before I get convinced with the results presented and consequently to accept the paper I need a response to a few important issues."*
(ii) Response: Thank you for this accurate summarization.

(i) RC2 comment 1:
*"Construction in China is very different from other regions. So the comparison with other regions is very difficult to accept. "*
(ii) Response: This concern is shared by RC1 as well. This part was added to respond the comments of one reviewer in the previous reviewing process. And we do notice the difficulty to conduct such comparisons, given the difference in building types across countries and variation in fragility analyses methods. Such a grand topic actually deserves individual deep-going study work in the future, and it is not enough to attach it as one section of our current work. Therefore, to narrow down the focus of this work, we decide to remove the comparisons in Section 4.2 and delete the related descriptions in the main context and the figures in the Appendix part.

(i) RC2 comment 2:
*"Only two Types RC and Masonry (sub-divided into two sub classes) are considered. EMS-98 has many more! "*
(ii) Response: As explained in Page 6, Line 11-18, compared with other building types, masonry and RC are more widely distributed and have relatively more fragility data. In the initial fragility data collection work, we actually collected empirical fragility data for more building types, including soil-wood, brick-wood, brick-concrete, RC, industrial frame, stone-wood, chuandou-timber, wood, stone and soil (these data are also available from the online supplement). Since another focus of our work is trying to develop intensity-PGA relationship by using fragility as the bridge, but for analytical fragility data, they are only available for masonry and RC building types. That's why finally only masonry and RC are considered.

(i) RC2 comment 3:

*"The final proposal (TableB2) for each Intensity gets an interval 1:2 and no results for I>X. Other studies show similar intervals but with centre value slightly deviated."*

(ii) Response: Thank you for your careful check. The values in Table B2 are the same as that in Chinese Official Seismic Intensity Scale (GB/T 17742-2008, in Chinese), which was issued in 2008 and was modified based on the old version GB/T 17742-1999 that issued in 1999. Maybe other studies you refer to were using the old values in GB/T 17742-1999?

(i) RC2 comment 4:

*"Errors on establishing the D1 to D4 or D5 are full of uncertainties and so I do not know if bringing the fragilities into place is reducing the error."*

(ii) Response: The categorization of damage of buildings into different classes is helpful for later-on risk analysis (i.e. from D1 to D5). On way to reduce the uncertainty in establishing the damage states is to increase the damage state/class used, but this will also increase the difficulty in applicating them during post-earthquake field investigation. Thus, trade-off needs to be made and currently many countries have five damage classes used.

When using fragility to represent the building damage, it provides a practical way to numerically compare fragilities derived from different methods. And the fragility curve can be attached with uncertainty of different damage states as well.

(i) RC2 comment 5:

*"The Indicator with people and bicycles is very interesting especially for China as there is much traffic with bikes. For other countries unfortunately, the situation is still too far!"*

(ii) Response: Exactly. That's why the comparison of Chinese building fragility with other countries are difficult to perform and should be separated from this work as an individual research.

(i) RC2 comment 6:

*"There should be a Figure representing graphically the results in Table B2."*

(ii) Response: Accepted. The following figure derived from Table B2 will be added to the Appendix part after Table B2.

[Figure]

Figure R2: The suggested correspondence relation between intensity and PGA/PGV range by Chinese Official Seismic Intensity Scale (GB/T 17742-2008, as explained in Table B2).

(i) RC2 comment 7:

*"Formula (5) is very old in California".*

(ii) Response: We're aware that currently the bilinear function is used to regress the relation between PGA and intensity, based on a rich bunch of PGA/PGV and intensity data (e.g. Worden et al., 2012). However, in our work, to develop the relation between intensity and PGA by using fragility as the bridge, the regression relation between ln(PGA) and intensity should be linear, not bilinear, as explained in Page 11 Line 35-Line 8 (Page 12) of the manuscript.

(i) RC2 comment 8:

*"Fig 3 and 4 should have line to be more understandable."*

(ii) Response: For empirical fragility data in Fig. 3, they are digitalized from post-earthquake surveys and the original data are also discrete without a line. For analytical fragility data in Fig. 4, they are also digitalized from individual papers or theses for PGA levels like 0.1g, 0.2g, 0.3g..., which are also discrete. Although the original analytical fragility curves are continuous, but not all the literature we scrutinized provide the formula of their fragility curves. Therefore, all the fragility data in Fig. 3 and 4 are discretely digitalized from each literature, thus it is difficult to add lines to these data when putting them together. That's why the following box-plot method will be applied to remove the outliers and find the median fragility value for each intensity/PGA level, and then to derive the most representative fragility curve for each damage state of each building type.

(i) RC2 comment 9:

*"Fig 5 and 6 should have line and same color."*

(ii) Response: For typical box-plot method, there is actually no line attached (regretfully it is also not common to do so). And the change of the building fragility with intensity/PGA level can be estimated from the median fragility value (as indicated by the red line and the blue dot within each box of Fig. 5 and Fig. 6, respectively). And later on, the median fragility curve will be further plotted in Fig. 7 and Fig. 8.

(i) RC2 comment 10:

*"Fig A1 could be merged with Fig 7 and Fig A2 with Fig 8."*

(ii) Response: Thank you for this good idea. The merged Fig. 7 (with Fig. A1) and Fig. 8 (Fig. A2) are as follows and will replace the old Fig. 7 and Fig. 8 in the revised manuscript.

[Figure]

Figure R3: Modified Fig. 7, merged with Fig. A1 in the Appendix.

[Figure]

Figure R4: Modified Figure 8, merged with Fig. A2 in the Appendix.

(i) RC2 comment 11:

*"Figs A6 and A7 are impossible to read!"*

(ii) Response: Sorry for this inconvenience when printing it out. Since the comparison of Chinese building fragility with that in other countries is a quite complex issue and deserves to be studied individually (as explained in above response), we decide to remove this part from this current manuscript. Thus, Fig. A5, A6, A7 will also be removed.

(i) RC2 comment 12:

*"Title could be slighted changed, inverting the order of ideas."*

(ii) Response:  Thank you for this proposal. However, first of all, the review of building fragility has been considered as a challenging task, since different approaches and methodologies are spread across scientific journals, conference proceedings, technical reports and software manuals, hindering the creation of an integrated framework that could allow the visualization, acquisition and comparison between all the existing curves. Therefore, the first focus of our work is to conduct such a review. Secondly, the derivation of intensity-PGA relation by using fragility as the bridge remains to be a tentative approach and uncertainties in between are difficult to fully handle, although we've tried hard to develop the uncertainty transmission

methodology in Appendix C. Thus, we prefer to mention this part as an implication from the review of building fragility. As such, we consider it's more appropriate to keep the current title as it is.

(i) RC2 comment 13:
*"I found several inconsistencies in the Reference List Attached Doc (References Corrections)."*
(ii) Response: Thank you very much for your careful check! We'll modify them in the revised version.

**Reference:**

Worden, C. B., Gerstenberger, M. C., Rhoades, D. A. and Wald, D. J.: Probabilistic Relationships between Ground-Motion Parameters and Modified Mercalli Intensity in California, Bulletin of the Seismological Society of America, 102(1), 204–221, doi:10.1785/0120110156, 2012.

---

## Author Response (AR1)

The authors are grateful to the time spent by Editor Maria Ana Baptista on monitoring the review process and by Prof. Mustafa Erdik (hereafter referred to as "RC1") and another anonymous reviewer (hereafter referred to as "RC2") on carefully reading our work and giving a series of constructive comments. All your efforts have greatly improved the manuscript. Point-to-point responses to reviewer comments are given as follows. The comments are in purple. Our responses are in blue. Accepted changes made in the revised manuscript are in green. The responses are followed by a marked-up version tracking all the changes we made in the revised manuscript. The authors are glad to respond to any of your further concern regarding this revised version.

**Part 1: Comments from RC1 and our Responses and Changes in the revised manuscript.**

(i) **General comments from RC1:** "The paper essentially aims the development of intensity-PGA relationships using a novel method that relies on comparison of intensity-based empirical and PGA-based analytical fragility relationships for the same building types from China. To fulfill this object, the authors first review the empirical building fragility database, mostly for China, scrutinize the data and derive the median Chinese intensity-based fragility relationships with basic treatment of uncertainties. For this empirical fragility study, three types of masonry buildings with different construction practices are considered. Secondly, the authors inspected publications that provide PGA-based analytical fragility functions (dependent on PGA) for the same damage classes and building categories. Thus, a solid fragility database and median fragility relationships, based on both intensity and PGA, are established for mainland China. For the derivation of median fragility relationships, the lognormal distribution is used with excellent goodness of fit. The paper culminates with the description and application of the novel approach for the development of the intensity-PGA relation by using fragility as the transfer medium. The results obtained are very valuable and compare well with limited relationships based on direct regression of measured PGA with the assessed intensity values."

(ii) Response: Thank you for this summarization and such positive judgement on our work.

(i) **RC1 comment 1:** "Very comprehensive literature review and description of ingredients and methodology on the assessment of fragility relationships from empirical data."

(ii) Response: Thank you for saying so.

(i) **RC1 comment 2:** "Text can be shortened since involves several repetitions of the objectives and methodologies."

(ii) Response: From the feedbacks of previous reviewing process, some reviewers misunderstood or partially neglected the focuses of this work. Therefore, although the objectives and methodologies are firstly mentioned in the Introduction section, to emphasize and to avoid possible misunderstanding from future readers, the focuses are thus reiterated in Section 4 and Section 5.2.

(iii) Changes in the revised manuscript: Accepted. The repetition in Section 4.2 is deleted.

(i) **RC1 comment 3:** "With the exception of the information provided on general approaches on the derivation of analytical vulnerabilities, not much detail is provided on the papers that the PGA-based analytical fragilities for the Chinese building stock. It appears that, with the exception of outlier removal, results on all these papers are given the same weight for the median fragility assessment. "

(ii) Response: The analytical fragility related studies for the Chinese building stock generally follow the classical methods and procedures as summarized in Page 4 Line27-36. Therefore, the details of these procedures are not presented in detail, since there is marginal exception.

You are right in that we do give the same weight for different intensity/PGA levels in regressing the median fragility curves, to avoid the introducing of extra subjective uncertainty. Since we noticed in previous literature, some researchers gave higher weight to lower intensity/PGA levels (e.g. Ding et al., 2017) in their regression, while others gave higher weight to higher intensity/PGA levels (e.g. Ma et al., 2014) with different focuses.

(i) **RC1 comment 4:** "The uncertainties in the fragility assessments are not adequately covered in the paper with the exception of uncertainties illustrated in Appendix Fig. A1-A4 and Table B1)."

(ii) Response: Thank you for pointing this out. RC2 also gives a detailed suggestion in regard of this. We'll accept the suggestions of you two and combine the error-bar analysis in the Appendix Fig. A1 and Fig. A2 with the median fragility curve in Fig. 7 and Fig. 8 in the main context, respectively.

(iii) Changes in the revised manuscript: Accepted. Fig. 7 and Fig. 8 are updated by integrating the error-bar analysis in Appendix Fig. A1 and Fig. A2, respectively. The original Fig. A1 and Fig. A2 are replaced by new figures, as explained below.

(i) **RC1 comment 5:** "Direct comparison of different fragility relationships is a difficult issue due to different building, damage state and ground motion intensity definitions and attributes considered in these relationships. This fact also manifests itself in this paper. Upon comparison of fragility relationships obtained in this paper with the results of several relevant international projects, only one (HAZUS Project) similarity for "Masonry – A" building type was found."

(ii) Response: Yes, this part was added to respond to the comments of one reviewer in the previous reviewing process. And we do notice the difficulty to conduct such comparison, given the difficulties as you summarized above. Actually, such a grand topic deserves individual deep-going study. Therefore, to keep the integrity and narrow down the focus of our current work, we'll remove the comparisons in Section 4.2 and delete the related descriptions in the main context and figures in the Appendix.

(iii) Changes in the revised manuscript: Section 4.2 and related descriptions in the context are removed. Figure A5, A6, A7 in the Appendix section are also deleted.

(i) **RC1 comment 6:** "The intensity-PGA relationships developed by using the correspondence between the empirical and PGA-based analytical fragility relationships is based on a novel approach and would be very valuable for use in international projects. However, a description on the relationship between the Chinese Official Seismic Intensity Scale (GB17742) and the other internationally adopted scales (e.g. MMI, MSK, EMS) may need to be included (or referenced) in the paper."

(ii) Response: Thank you for this suggestion. We found in previous studies (e.g. in Daniell, 2014) such work has been conducted. We'll add these references in the main context and add the following graphic comparison of different intensity scales into the Appendix.

(iii) Changes in the revised manuscript: The following Figure R1 is added into the Appendix as the new Figure A1. Related references are also added in the main context.

[Figure]

Figure R1: Comparison of Intensity Scales in Daniell (2014, in his Figure 9), after the work of Gorshkov and Shenkareva (1960), Barosh (1969), Musson et al. (2010) (Note: in this figure, "Liedu-1980/1999" represents the Chinese Seismic Intensity Scale).

(i) **RC1 comment 7:** "The methodology in the transmission of uncertainty from empirical/analytical fragility database to the intensity-PGA relation is provided in Appendix C. This transmission of uncertainty is important and should preferably be integrated into the main text of the paper."

(ii) Response: Thank you for this suggestion. But since the uncertainty of this newly derived intensity-PGA relationship is mentioned only as a number in Section 5.4, to keep the structure of this work clear and also to narrow down the focus, we still consider that it is better to put the uncertainty transmission methodology in the Appendix part for interested readers to have a further check.

**Part 2: Comments from RC2 and our Responses and Changes in the revised manuscript.**

(i) **General comments from RC2:** "This is a very interesting paper tying the fragility curves to vulnerability and through this get a better correlation into conversion of Intensities to PGA. The authors reviewed a large number of data published on Chinese earthquakes since 1975 for two main typologies: RC and Masonry: 69 in terms of Chinese Intensity Scale and 18 on PGA. They used fragility curves obtained from experimental (in function of INT) and from numerical developments (in function of PGA) to obtain a conversion between INT vs PGA. To expand the results obtained to other places around the World, and not only to the Chinese construction, in a tentative to propose a world-wide conversion, they compare the results with other proposals in Europe and US (PERPETUATE; SHARE; GEM; etc.). But before I get convinced with the results presented and consequently to accept the paper I need a response to a few important issues."

(ii) Response: Thank you for your interests and this accurate summarization.

(i) **RC2 comment 1:** "Construction in China is very different from other regions. So the comparison with other regions is very difficult to accept."

(ii) Response: This concern is shared by RC1 as well. The comparison with other international projects with similar focus was added to respond to the comments of one reviewer in the previous reviewing process. Actually, it is a painstaking process to conduct such comparison, since the difference in building types across countries and variation in fragility analysis methods pose great difficulty in doing so. Although a lot of efforts have been made, only one (HAZUS Project) similarity for "Masonry_A" building type was found. Such a grand topic deserves an individual and deep-going study in the future, and it is far from enough to attach it as one section of this current work. Therefore, we decide to remove the comparisons in Section 4.2 and delete the related descriptions in the main context and the figures in the Appendix part.

(iii) Changes in the revised manuscript: Section 4.2 and related descriptions in the context are removed. Figure A5, A6, A7 in the Appendix section are also deleted.

(i) **RC2 comment 2:** "Only two Types RC and Masonry (sub-divided into two sub classes) are considered. EMS-98 has many more!"

(ii) Response: Thank you for pointing this out. In the initial fragility data collection work, we actually collected empirical fragility data for more building types, including soil-wood, brick-wood, brick-concrete, RC, industrial frame, stone-wood, chuandou-timber, wood, stone and soil (these data are also available from the online supplement). As explained in Page 6, Line 11-18, since another focus of our work is trying to develop intensity-PGA relationship by using fragility as the bridge, but for analytical fragility data, they are only available for masonry and RC building types. Compared with other building types, masonry and RC are more widely distributed and have relatively more fragility data. That's why only masonry and RC are used. Due to the lack or incompleteness of building seismic resistance information in some reviewed literature and also to ensure data abundancy, it is subjectively decided to divide masonry/RC into two sub-classes.

(i) **RC2 comment 3:** "The final proposal (TableB2) for each Intensity gets an interval 1:2 and no results for I>X. Other studies show similar intervals but with centre value slightly deviated."

(ii) Response: Thank you for your careful check. We further checked the values in Table B2 and they are the same as those in Chinese Seismic Intensity Scale (GB/T 17742-2008, in Chinese). Compared with previous versions (GB/T 17742-1980, GB/T 17742-1999), the current version (GB/T 17742-2008) provides more accurate

description of reference building type used to derive intensity. The average damage index also changes accordingly. But the correspondence relation between intensity and PGA/PGV are the same in all these three officially issued intensity scales.

(i) **RC2 comment 4:** "Errors on establishing the D1 to D4 or D5 are full of uncertainties and so I do not know if bringing the fragilities into place is reducing the error."

(ii) Response: We agree that assigning building damage to different damage states (e.g. from D1 to D4 or D5) is of varying uncertainty. And one way to reduce such uncertainty could be to introduce more damage states/classes. However, one main application of the assignment of building damage into different states/classes is for later-on risk analysis. The increase of damage states will inevitably enhance the difficulty and complexity in applicating them in post-earthquake field investigations. Therefore, trade-off needs to be made and that may be why currently many countries have five damage classes under use.

From the definition of building fragility, which describes the probability to exceed each damage state at various ground shaking levels. The derivation of building fragility curve (as indicated by Eq. (1)-(2)) is dependent on the pre-definition of each damage state indicator. Therefore, we consider that building fragility only loyally inherits the uncertainty from defining different damage states but is not able to reduce such uncertainty.

The advantage of introducing building fragility is that, it provides a practical way to numerically compare fragilities derived from different empirical/analytical methods and fragility curves can also be attached with uncertainties of different damage states.

(i) **RC2 comment 5**: "The Indicator with people and bicycles is very interesting especially for China as there is much traffic with bikes. For other countries unfortunately, the situation is still too far!"

(ii) Response: Exactly. That's further indicates the difficulty in conducting the comparison of Chinese building fragility with other countries.

(i) **RC2 comment 6**: "There should be a Figure representing graphically the results in Table B2."

(ii) Response: Thank you for this good advice. The following figure indicating the correspondence relation between intensity and PGA/PGV (available for intensity V to X) in Table B2 will be added.

(iii) Changes in the revised manuscript: Accepted. The following Figure R2 is added into the Appendix section as the new Fig. A2.

[Figure]

Figure R2: The suggested correspondence relation between intensity and PGA/PGV range by Chinese Official Seismic Intensity Scale (GB/T 17742-2008, as listed in Table B2).

(i) **RC2 comment 7:** "Formula (5) is very old in California".

(ii) Response: We're aware of the current trend that the bilinear function is used to regress the relation between PGA and intensity, based on a rich bunch of observational PGA/PGV and intensity data (e.g. Worden et al., 2012). However, in our work, to develop the relation between intensity and PGA by using fragility as the bridge, the regression relation between ln(PGA) and intensity should be linear, as explained in Page 11 Line 35-Line 8 (Page 12) and derived by Eq. (4).

(i) **RC2 comment 8:** "Fig 3 and 4 should have line to be more understandable."

(ii) Response: For empirical fragility data in Fig. 3, they are digitalized from many post-earthquake surveys and thus are difficult to be represented by one single line.

For analytical fragility data in Fig. 4, they are also digitalized from individual papers or theses for PGA levels like 0.1g, 0.2g, 0.3g…, which are also discrete. Although the original analytical fragility curves are continuous, but not all the literature we scrutinized provided the formula of their fragility curves.

Therefore, all the fragility data in Fig. 3 and 4 are discretely digitalized from different literature, and it is difficult to add lines to these data when putting them together. That's why the following box-plot method will be applied to remove the outliers and find the median fragility for each intensity/PGA level.

(i) **RC2 comment 9:** "Fig 5 and 6 should have line and same color."

(ii) Response: For typical box-plot method, there is actually no line attached. Regretfully, it is also not common to do so. The change of the building fragility with intensity/PGA level can be estimated from the median fragility value (as indicated by the red line and the blue dot within each box of Fig. 5 and Fig. 6, respectively). And the median fragility curve will be represented by lines in Fig. 7 and Fig. 8. But we will adopt your advice to replot Fig. 6.

(iii) Changes in the revised manuscript: Partially accepted. Fig. 6 is updated by the following Figure R3.

[Figure]

Figure R3: Replotted Figure 6 to keep the same color as Figure 5 in the main context.

(i) **RC2 comment 10:** "Fig A1 could be merged with Fig 7 and Fig A2 with Fig 8."

(ii) Response: Thank you for this good idea. We'll do this.

(iii) Changes in the revised manuscript: Accepted. Fig. 7 and Fig. 8 are updated by the following Fig. R4 and R5, in which the median fragility curve is combined with the error-bar analysis in Appendix Fig. A1 and Fig. A2, respectively. The original Fig. A1 and Fig. A2 are removed.

[Figure]

Figure R4: Updated Fig. 7, merged with original Fig. A1 in the Appendix.

[Figure]

Figure R5: Updated Figure 8, merged with original Fig. A2 in the Appendix.

(i) **RC2 comment 11:** "Figs A6 and A7 are impossible to read!"

(ii) Response: Sorry for this inconvenience. Since the comparison of Chinese building fragility with that in other countries is a quite complex issue and deserves to be studied individually (as explained in above responses), we decide to remove this part from this current manuscript. Thus, Fig. A5, A6, A7 will also be removed.

(iii) Changes in the revised manuscript: Based on the responses to previous comments, Fig. A5-A7 are removed.

(i) **RC2 comment 12:** "Title could be slighted changed, inverting the order of ideas."

(ii) Response: Thank you for this proposal. However, first of all, the review of building fragility has been considered as a challenging task, since different approaches and methodologies are spread across scientific journals, conference proceedings, technical reports and software manuals, hindering the creation of an integrated framework that could allow the visualization, acquisition and comparison between all the existing curves. Therefore, the first focus of our work is to conduct such a review. Secondly, the derivation of intensity-PGA relation by using fragility as the bridge remains to be a tentative approach and uncertainties in between are difficult to fully handle, although we've tried hard to develop the uncertainty transmission methodology in Appendix C. In this regard, we think it is more appropriate to mention this part as an implication from the review of building fragility.

(iii) Changes in the revised manuscript: Given the above consideration, it is more appropriate to keep the title as it is.

(i) **RC2 comment 13:** "I found several inconsistencies in the Reference List Attached Doc (References Corrections)."

(ii) Response: Thank you very much for your careful check!

(iii) Changes in the revised manuscript: Accepted. Such inconsistencies in the revised version are rectified.

**Reference:**

[revised manuscript text omitted]